# Antibacterial effects of nanopillar surfaces are mediated by cell impedance, penetration and induction of oxidative stress

J. Jenkins [1], J. Mantell[2], C. Neal[2], A. Gholinia [3], P. Verkade [2], A.H. Nobbs [1✉] & B. Su[1✉]

Some insects, such as dragonflies, have evolved nanoprotrusions on their wings that rupture bacteria on contact. This has inspired the design of antibacterial implant surfaces with insect-wing mimetic nanopillars made of synthetic materials. Here, we characterise the physiological and morphological effects of mimetic titanium nanopillars on bacteria. The nanopillars induce deformation and penetration of the Gram-positive and Gram-negative bacterial cell envelope, but do not rupture or lyse bacteria. They can also inhibit bacterial cell division, and trigger production of reactive oxygen species and increased abundance of oxidative stress proteins. Our results indicate that nanopillars' antibacterial activities may be mediated by oxidative stress, and do not necessarily require bacterial lysis.

[1] Bristol Dental School, University of Bristol, Bristol, UK. [2] School of Biochemistry, University of Bristol, Bristol, UK. [3] School of Materials Science, University of Manchester, Manchester, UK. ✉email: angela.nobbs@bristol.ac.uk; b.su@bristol.ac.uk

It is now well established that insect wings, including the cicada and dragonfly, possess antibacterial and antifungal properties. Previous studies indicate that this process is mediated by the physical nanoprotrusions found on the wing surface, which ultimately stretch and damage the microbial cell upon contact, leading to lysis and death[1–6]. This effect was first observed in *Pseudomonas aeruginosa*, where individual bacteria were shown to sink and spread between the nanopillars found on *Psaltoda claripennis* (cicada) wings, occurring in attachment/killing cycles of 20 min (ref. [1]); this mechanism was proposed to occur by mechanical rupture of the cell. Bactericidal effects were also described for other Gram-negative bacteria, including *Branhamella catarrhalis*, *Escherichia coli* and *Pseudomonas fluorescens*[6]. Alongside cicada, dragonfly wings have been shown to possess efficient bactericidal properties. *Diplacodes bipunctata* wings were found to mediate killing of both Gram-negative (*P. aeruginosa*) and Gram-positive (*Staphylococcus aureus* and *Bacillus subtilis*) bacteria. The capillary architecture of *D. bipunctata* wing nanoprotrusions is hypothesised to enhance cell wall stress and deformation, thereby extending bactericidal activity towards Gram-positive cell types[2].

The unique bactericidal properties of cicada and dragonfly wings have drawn significant research interest[7–10], as the physical nature of bacterial killing could provide an effective strategy to prevent biofilm formation, and infection of indwelling and implantable devices, while negating the current need to use materials impregnated with antibiotics. To date, a wide range of nanofabrication techniques have been utilised to generate bactericidal nanotopographies on synthetic materials, including black silicon (bSi)[2], titanium[11], titanium alloy[12] and polymers[13].

Several models for the process of contact killing have been proposed. The biophysical model suggested that cicada wing nanopillars induce physical stretching of the cell membrane upon contact, leading to bacterial rupture and death, and cell rigidity was reported to be an important determinant of susceptibility[14]. In support of this, the elastic mechanical model proposed that Gram-positive bacteria are less susceptible to nanopillar deformation and rupture owing to their lower maximum stretching capacity. However, by increasing nanopillar sharpness and spacing, the antibacterial properties may be enhanced[15]. A quantitative thermodynamic model proposed that the bactericidal activity of nanopatterned surfaces is directly related to the balance between adhesion energy and deformation energy. The model predicts that within certain dimensions (nanopillar radius 0–50 nm and nanopillar spacing 100–250 nm), the stretching degree applied across the bacterial envelope is enhanced by nanoarrays with greater nanopillar diameters (50 nm) and reduced nanopillar spacing (100 nm)[16].

Multiple factors are reported to influence the antimicrobial efficacy of natural and synthetic nanopillars. Of note, the microbial adhesion force to a nanotopography has been shown to directly influence viability. *Saccharomyces cerevisiae* rupturing was greatest in strains that adhered most strongly to cicada wing nanopillars[5]. Similarly, the strong adhesion between dragonfly nanopillars and *E. coli* extracellular polymeric substance is reported to promote bacterial membrane damage[4]. The rigidity of bacterial cells has also been found to influence their susceptibility to mechanical rupture[6,14], whereby Gram-negative bacteria were more sensitive to nanopillar-mediated stretching. This observation most likely reflects variations in envelope architecture. Gram-negative envelopes consist of an outer and inner membrane, separated by a thin ($\approx$5 nm) peptidoglycan cell wall that occupies the periplasmic space. By contrast, the Gram-positive bacterial cell wall is significantly thicker ($\approx$20–100 nm)[13,17], which may reduce stretching sensitivity. In addition to this, nanotopography geometries, including aspect ratio and nanopillar density, have been shown to influence the efficiency of bactericidal activity[3,5,13,18].

Although the bactericidal activity of natural and synthetic nanopillars has been widely reported, no consensus has been reached on the precise mechanism that leads to the microbial cell death and importantly, while many studies infer that nanopillars mediate mechanical rupture of bacterial cells, this has not been shown conclusively. In addition to their capacity to rupture and lyse bacteria, nanostructured surfaces are reported to alter the genomic and proteomic profile of bacteria. Of note, type-1 fimbriae expression was significantly downregulated in *E. coli* adhered to gold nanoparticle surfaces, while stress response proteins involved in protection from DNA and membrane damage had been upregulated[19,20]. Consistent with these findings, *E. coli* incubated on single-walled carbon nanotubes (CNTs) expressed high levels of stress response proteins related to cell membrane damage and oxidative stress[21]. Physiological responses of this nature may contribute significantly to the overall bactericidal activity of nanostructured surfaces. Elucidating these molecular changes is therefore essential to progress our understanding of the mechanistic processes that lead to bacterial cell death.

To better understand and validate the antimicrobial mechanisms relating to specific nanotopographies, this study set out to determine the morphological and physiological responses of bacteria to dragonfly mimetic nanopillars, as these nanotopographies have been reported to possess highly efficient bactericidal activity against both Gram-negative and Gram-positive bacteria[2,4]. To elucidate the impact of titanium dioxide ($TiO_2$) nanopillars on bacterial envelope integrity, and to determine whether nanopillars induce mechanical rupture and cell lysis, electron tomography techniques were utilised to reconstruct detailed 3D visualisations of bacteria adhered to nanopillars. To gain a deeper insight into the molecular changes that occur on nanopillar surfaces, changes in protein expression were investigated by quantitative tandem mass tagging (TMT) proteomic analysis. Bacterial cell viability (BacTiter-Glo, RealTime-Glo) assays were additionally employed to quantify the growth and viability of *S. aureus*, *E. coli* and *Klebsiella pneumoniae* populations incubated with $TiO_2$ nanopillars.

## Results

**Characterisation of $TiO_2$ nanopillars on titanium substrate**. A thermal oxidation technique[12,22] was used to generate nanopillars on grade 5 titanium alloy (Ti-6Al-4V), a material widely used for orthopaedic implants[23]. Nanopillar surface NW-850-5 was oxidised at 850 °C for 5 min (Fig. 1), generating a nanotopography closely mimicking the nanoprotrusions found on dragonfly wings. Nanopillars consisted of $TiO_2$ and were predominantly comprised of rutile $TiO_2$ (98%; Supplementary Figs. 1–2).

**$TiO_2$ nanopillars induce envelope deformation and penetration**. The bactericidal action of cicada wing nanopillars was reported to occur immediately after seeding with *P. aeruginosa*[1]. Furthermore, the killing rates observed on dragonfly wing mimetic bSi were highest within 3 h (ref. [2]). These conclusions were partly drawn from qualitative analysis, using scanning electron microscopy (SEM), which revealed significant changes in bacterial morphology. To enable a direct comparison with these studies, high-resolution SEM was used to determine whether $TiO_2$ nanopillars induced mechanical rupture and lysis of *S. aureus*, *E. coli* or *K. pneumoniae* after 3 h.

On control surfaces, bacteria exhibited the characteristic morphologies of coccoid (*S. aureus*) or bacillus (*E. coli* and *K. pneumoniae*) cells after 3-h incubations (Fig. 2). These cell

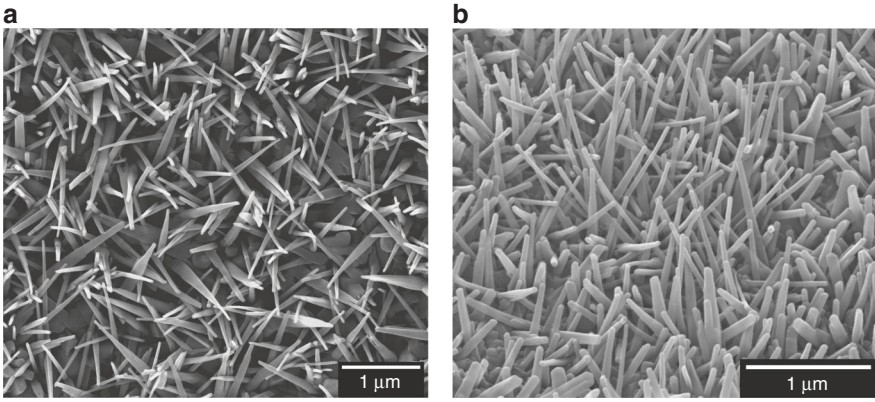

**Fig. 1 Characterisation of TiO₂ nanopillars.** Scanning electron micrographs of TiO₂ nanopillar surface NW-850-5 from a top view **a** and 40° stage tilt **b**. NW-850-5 was generated at 850 °C for a duration of 5 min. Scale bars: 1 μm. Micrographs are representative of three thermal oxidation batches ($n = 3$), each batch containing 25 surfaces.

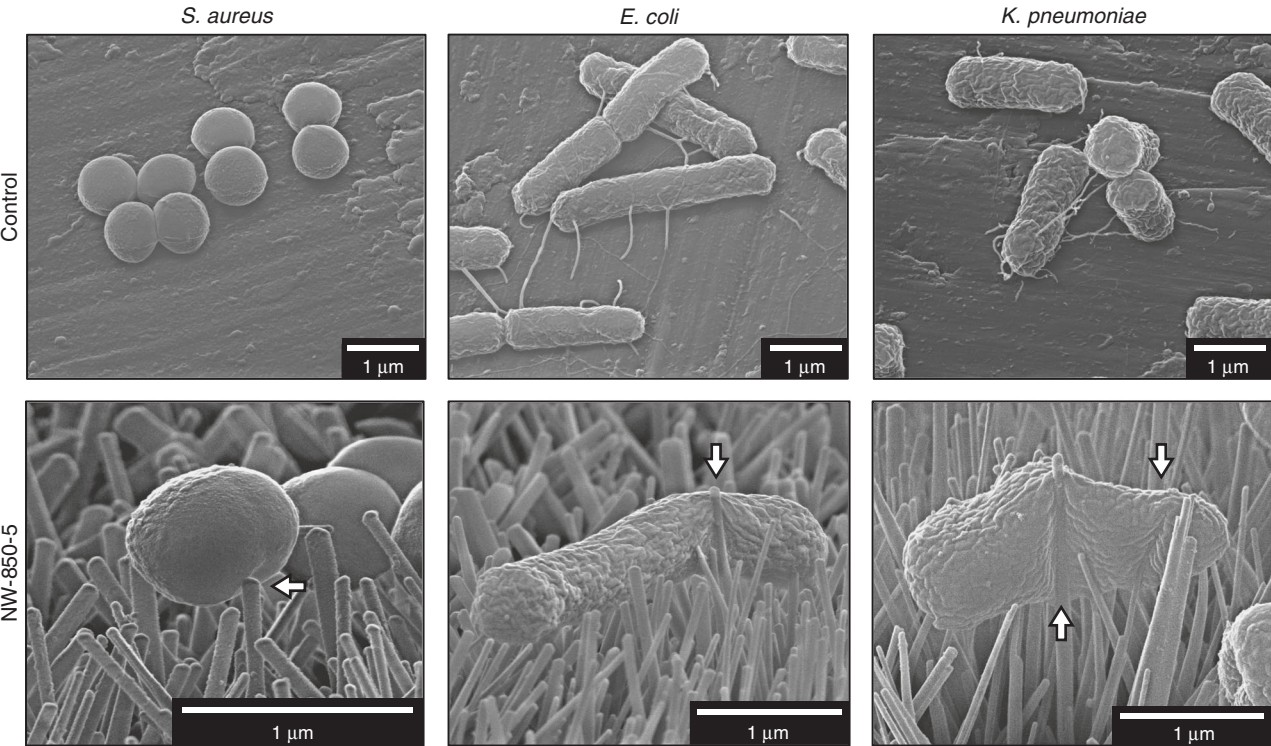

**Fig. 2 Determining bacterial morphology on nanopillar surface NW-850-5 using SEM.** Scanning electron micrographs of Gram-positive (*S. aureus*) or Gram-negative (*E. coli* and *K. pneumoniae*) bacteria following 3-h static incubation on flat titanium alloy (control) and TiO₂ nanopillar surface NW-850-5. White arrows highlight regions of nanopillar-induced envelope deformation. Micrographs are representative of three surfaces ($n = 3$), with each surface visualised in at least three different areas.

morphologies were also observed after 0.5-, 1- and 10-h incubations (Supplementary Figs. 3–5). The coverage of each bacterium increased on the control surfaces between 3 and 10 h, and there was extensive evidence of microcolony formation, indicative of early stage biofilm formation. In contrast, coverage and density were visibly lower for all bacteria on nanopillar surfaces (Supplementary Figs. 6–8). Overall, the morphologies of *S. aureus* incubated on NW-850-5 were comparable to bacteria on the controls but envelope deformation was observed, defined here as the process by which nanopillars indent or change the surface morphology of the bacterial envelope in a contact-dependent manner. Nanopillar-induced envelope deformation was also observed for *E. coli* and *K. pneumoniae* incubated on NW-850-5 (Fig. 2). Despite this, there was no evidence that

nanopillars had induced mechanical rupture or lysis of bacterial cells, defined here as the loss of cytosolic content and turgor pressure, resulting in bacteria sinking into the nanotopography. Additionally, the morphologies of bacteria after 0.5-, 1- and 10-h incubations on nanopillar surfaces were comparable to those of bacteria on control surfaces, with only subtle indications of nanopillar-induced envelope deformation for *E. coli* and *K. pneumoniae*. It was also noted that *E. coli* and *K. pneumoniae* microcolonies appeared to express fimbria-like surface appendages on control surfaces, while these structures were largely absent from bacteria in contact with the nanopillars.

The observation of nanopillar-induced envelope deformation for *S. aureus*, *E. coli* and *K. pneumoniae* was partly consistent with the proposed mechanism of nanopillar-mediated contact

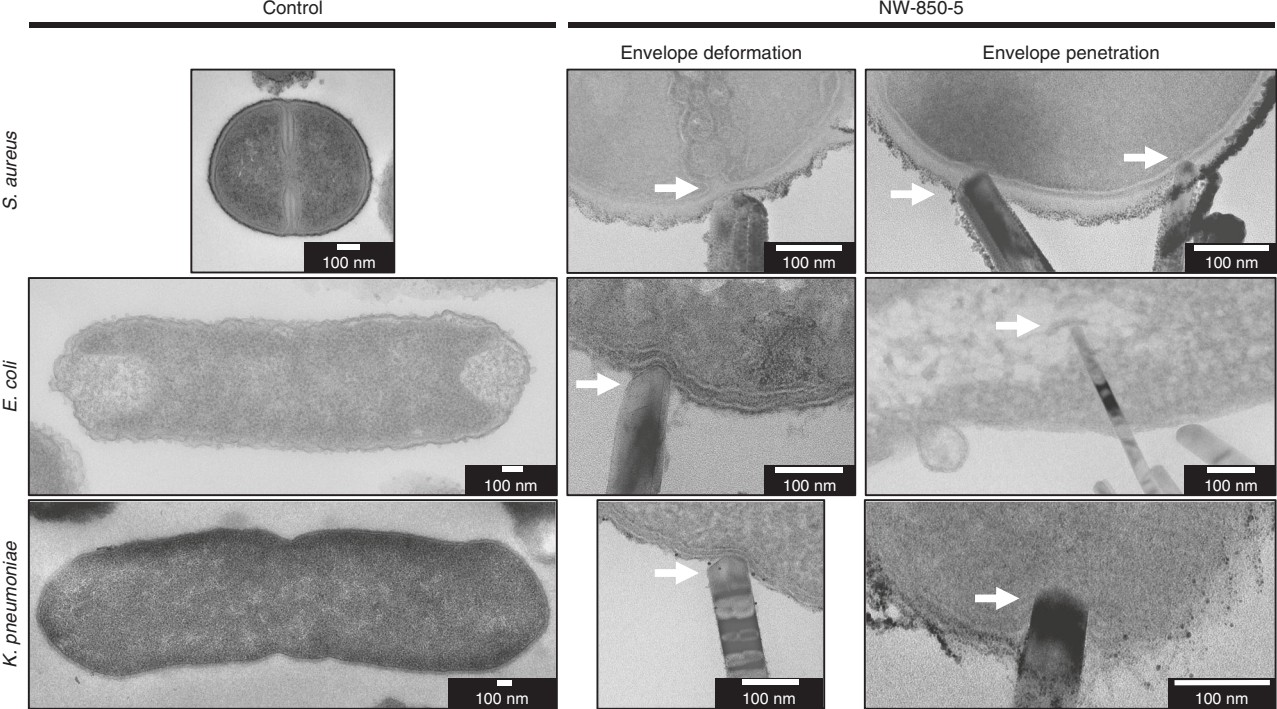

**Fig. 3 Nanopillar-induced envelope deformation and penetration in Gram-negative and Gram-positive bacteria.** TEM micrographs of *S. aureus*, *E. coli* and *K. pneumoniae* following incubation on control surfaces and NW-850-5 for 3 h. Regions where nanopillars had deformed or penetrated the bacterial envelope have been marked by white arrows. Cross section thickness: 70 nm. Micrographs are representative of three independent surfaces (*n* = 3).

killing, which suggests that as bacteria adhere to the nanopillar surface, the envelope is stretched to the point of physical rupture, leading to cell death[14]. However, it was unclear whether nanopillar-induced deformation could lead to the penetration of the bacterial envelope, defined as the capacity of nanopillars to pierce through the bacterial envelope, thereby disrupting the continuous barrier between the cytosol and extracellular environment. To investigate this in greater detail and determine whether penetration could induce cell lysis, electron microscopy techniques were utilised that could resolve the interface between nanopillars and the bacterial envelope, namely transmission electron microscopy (TEM) and focused ion beam SEM (FIB-SEM).

Following recovery from control surfaces, *S. aureus* displayed a uniform and continuous envelope and appeared to be dividing, as was evident from the bacterial septum (Fig. 3; top left). The characteristic architecture of a Gram-positive envelope was observed, with a plasma membrane (average: 8 nm ± 1 nm) surrounded by a peptidoglycan cell wall (average: 22 nm ± 2 nm). The envelopes of *E. coli* and *K. pneumoniae* were comparable, both consisting of outer (12 nm ± 2 nm and 9 nm ± 2 nm, respectively) and inner (9 nm ± 2 nm and 6 nm ± 1 nm, respectively) membranes, separated by a thin (≈5 nm) peptidoglycan layer (Fig. 3; middle and bottom left). *S. aureus* ultrastructure was largely unaffected by the interactions with NW-850-5, with envelope deformation and penetration observed at a low frequency (5%; Fig. 3; top middle and top right). In contrast, TEM analysis provided strong evidence that nanopillars had deformed the envelope of *E. coli* (Fig. 3; middle and middle bottom); indentations in close proximity to the nanopillar tips were observed in 26% of cells, while envelope penetration was observed at a frequency of 19%. For *K. pneumoniae*, nanopillars had induced envelope deformation in 11% of cells and had penetrated 8% of cells analysed.

To determine whether the envelope deformation was localised to the point of nanopillar contact, or whether global perturbations had occurred, tomography analysis was utilised to monitor changes in envelope structure and shape with distance across the cell. Tomography was performed on an *E. coli* cross section, measuring 250 nm in thickness. Initial TEM analysis had identified a single nanopillar interacting with the cell (Fig. 4a). Tomographic slices were extracted from the start, middle and point of nanopillar contact; these highlighted that the envelope structure was uniform at the beginning (Fig. 4b) and middle (Fig. 4c) of the tomogram, while deformation was only seen at point of nanopillar contact (Fig. 4d). This confirmed the localised nature of these interactions. Analysis of the 3D reconstruction of *E. coli* revealed that the bacterial envelope had been deformed by ~80–100 nm, but this had not resulted in mechanical rupture or lysis (Fig. 4e). Tomography was also performed on a single *K. pneumoniae* cell that appeared to be penetrated by multiple nanopillars (Fig. 5a). Analysis of multiple tomographic slices confirmed that five out of six nanopillars had penetrated, with depths varying from 103–179 nm (Fig. 5b).

To investigate the effects of nanopillars on bacterial envelope morphology further, cross sectional analysis of *E. coli* and *S. aureus* was performed using FIB-SEM. FIB-SEM is an effective tool for investigating cell–material interfaces with the ability to selectively analyse individual bacterial cells[24,25]. Preliminary SEM analysis revealed a single nanopillar (nanopillar 1) causing significant envelope deformation on one side of *E. coli*, leading to the bacterium being forced against a nanopillar (nanopillar 2) on the opposite side (Fig. 6). Automated ion beam milling was performed to reveal whether either nanopillar had penetrated the envelope. Although nanopillar 1 had deformed the bacterial envelope by 191 nm, it had not penetrated the cell. Nonetheless, it was evident from such analyses that nanopillars had capacity to act as physical barriers, entrapping bacterial cells and possibly impeding division on the horizontal plane.

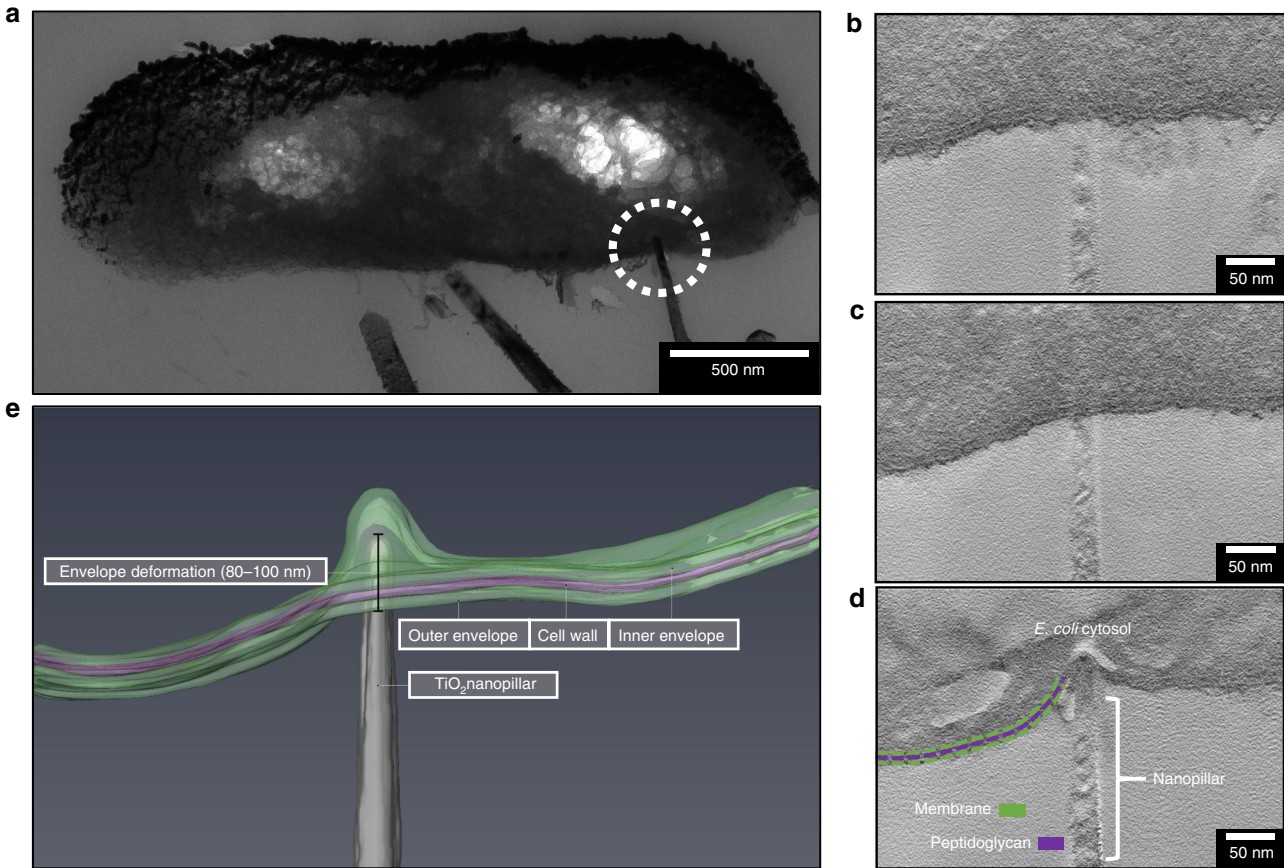

**Fig. 4 Tomogram reconstruction of nanopillar-induced envelope deformation in *E. coli*.** A tilt series was acquired of *E. coli* **a**, where a single nanopillar appeared to be penetrating the envelope (dashed white circle). Analysis of tomographic slices from the start **b**, middle **c** and point of nanopillar contact **d** revealed that the nanopillar had clearly indented the envelope but had not pierced through. The interaction was therefore categorised as nanopillar-induced envelope deformation, localised at the point of nanopillar contact. This interaction is highlighted in the 3D reconstruction shown in **e**. Electron tomography was performed on three *E. coli* cells interacting with TiO₂ nanopillars, these were acquired in two tilt series.

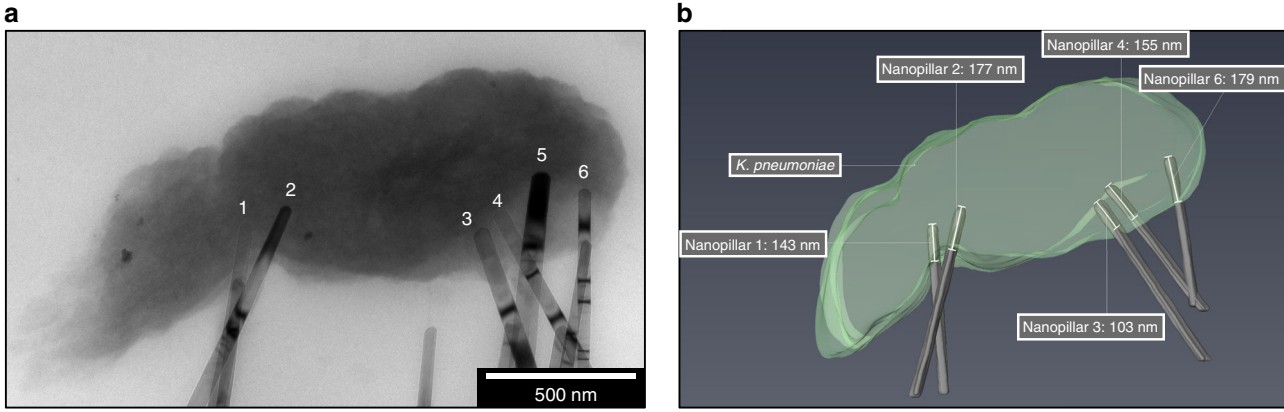

**Fig. 5 Tomogram reconstruction of nanopillar-induced envelope penetration in *K. pneumoniae*.** A tilt series was acquired of *K. pneumoniae* under bright-field TEM **a**, where multiple nanopillars appeared to be penetrating the envelope (1–6). Analysis of nanopillar tip coordinates in *x*, *y* and *z* revealed that nanopillars 1–4 and 6 had penetrated the envelope, while nanopillar 5 was only deforming the cell. A 3D reconstruction of the tomogram revealed all nanopillars had penetrated the bacterial envelope by at least 100 nm **b**. The dark bands seen within nanopillars are bend contours; these spatial contrasts arise from local bending or deformation of nanopillars. Electron tomography was performed on five *K. pneumoniae* cells interacting with TiO₂ nanopillars, these were acquired in four tilt series.

Cross sectional analysis was also performed on *S. aureus* in contact with three nanopillars (Fig. 7). Analysis of individual FIB cross sections confirmed that nanopillar 2 had penetrated the bacterial envelope by 48 nm, surpassing the envelope by ~15 nm. Thus, in contrast to *E. coli*, where nanopillar 1 had induced significant local deformation, in *S. aureus*, envelope penetration did not appear to affect cell morphology. These findings corroborate our cross sectional analysis using TEM, which provides evidence that nanopillars had penetrated the envelope of *S. aureus* without significant alterations to morphology.

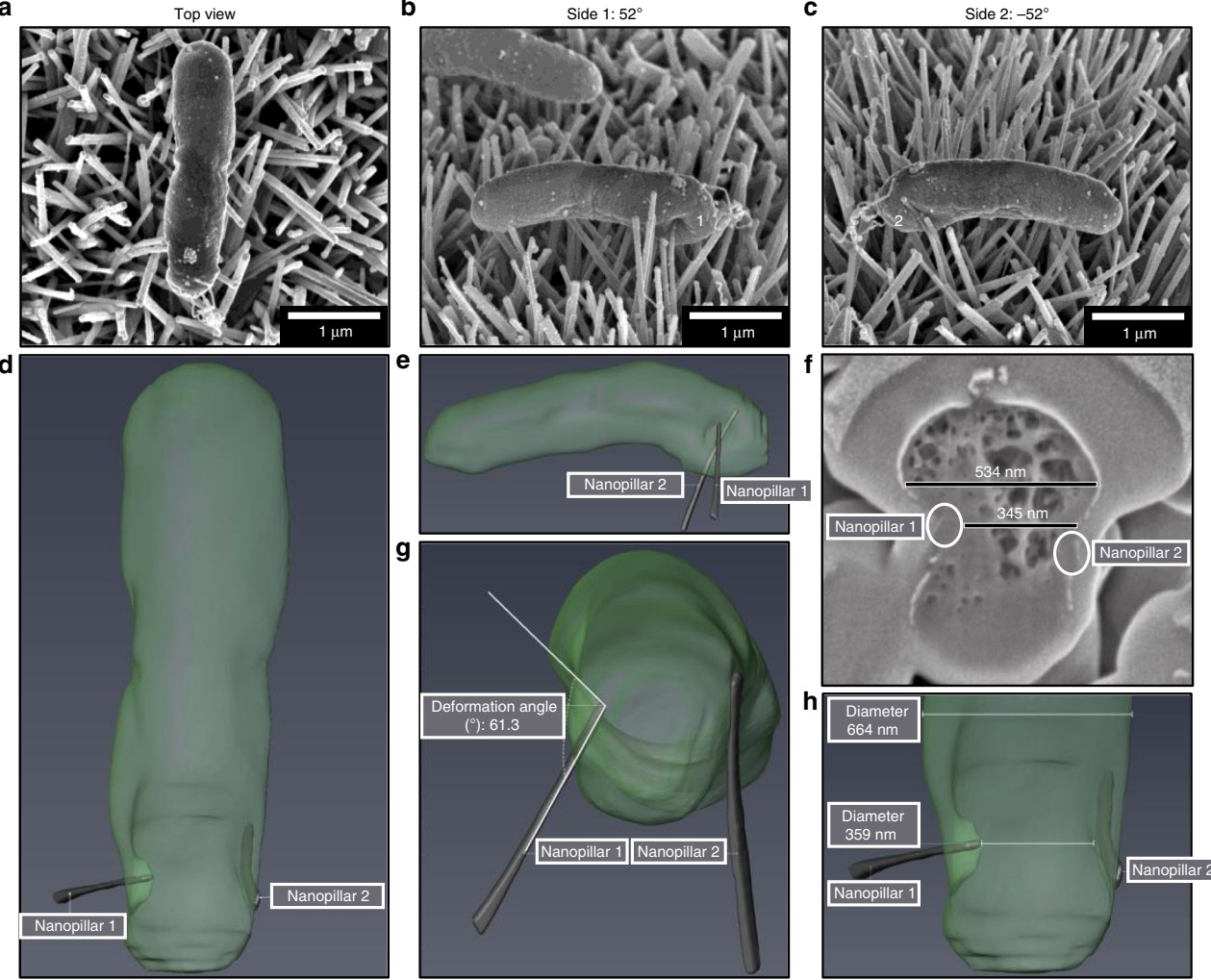

**Fig. 6 3D FIB-SEM reconstruction of *E. coli*.** Automated FIB-SEM cross sectional analysis was performed on an *E. coli* cell **a** that was pinned between two nanopillars **b**, **c**. The focused ion beam produced 120 cross sections (30 nm each) that were imaged, and reconstructed in Avizo **d**, **e**. Analysis of *E. coli* cross section #30 showed that nanopillar 1 had forced *E. coli* into nanopillar 2, resulting in a 189 nm reduction in cell width **f**. While nanopillar 1 had induced significant envelope deformation, no evidence of envelope penetration was found, and the width of *E. coli* cell quickly restored to normal either side of the contact point, indicating that the cell had not lost turgor pressure **g**, **h**. FIB-SEM analysis was performed on four *E. coli* cells interacting with TiO$_2$ nanopillars.

**TiO$_2$ nanopillars impair cell viability**. The observations of nanopillar-induced envelope deformation and penetration in both Gram-positive and Gram-negative bacteria, led us to consider that nanopillars may have affected cell viability. The antibacterial activity of NW-850-5 compared to control titanium surfaces was initially assessed using the luminescent metabolic indicator assay RealTime-Glo. Growth of *S. aureus*, *E. coli* or *K. pneumoniae* was monitored in situ by recording luminescence in a temperature-controlled (37 °C) plate reader for 10 h (Fig. 8). The rate of luminescence production by *S. aureus* was noticeably slower on NW-850-5 and plateaued after 6 h, while on controls, the signal increased over the entire 10-h incubation period. Levels of metabolic activity differed significantly from controls on NW-850-5 after 9 h (Fig. 8a). Similar trends were observed for *E. coli*, although the differences between nanopillar and control surfaces did not reach statistical significance (Fig. 8b). A contributory factor to this was that the absolute relative luminescence unit (RLU) values were significantly (1000-fold) lower for *E. coli* than for *S. aureus*, which most likely reflects the lower reducing capacity of Gram-negative bacteria. This was further supported when *K. pneumoniae* was tested. The absolute RLU values were much lower and no significant differences were observed, although the luminescence profile for NW-850-5 was distinct from the other two surfaces (Fig. 8c). Of note, *E. coli* and *K. pneumoniae* studies showed a reduction in luminescence signal on controls from 6 and 5 h, respectively. This may indicate that other factors, independent of the surface, were affecting growth at these times.

Given the low sensitivity of RealTime-Glo with Gram-negative bacteria in this study, viability testing was additionally performed using endpoint assay BacTiter-Glo, and luminescence data were converted to bacterial CFU based on standard curves (Supplementary Fig. 9). Contrary to RealTime-Glo analyses, BacTiter-Glo revealed significant differences in *E. coli* viability on NW-850-5 following 3- and 10-h incubations relative to controls. Similarly, BacTiter-Glo revealed significant differences in *K. pneumoniae* viability on NW-850-5 following 3-h incubations relative to controls; although significance was not reached at 10 h (Fig. 8c). For *S. aureus*, significant reductions in CFU relative to controls were observed on NW-850-5 following 10-h incubation (Fig. 8a), corroborating the RealTime-Glo data.

**Bacterial proteomic response to nanopillar surface NW-850-5**. SEM determination of bacterial densities (Supplementary Figs. 6–8)

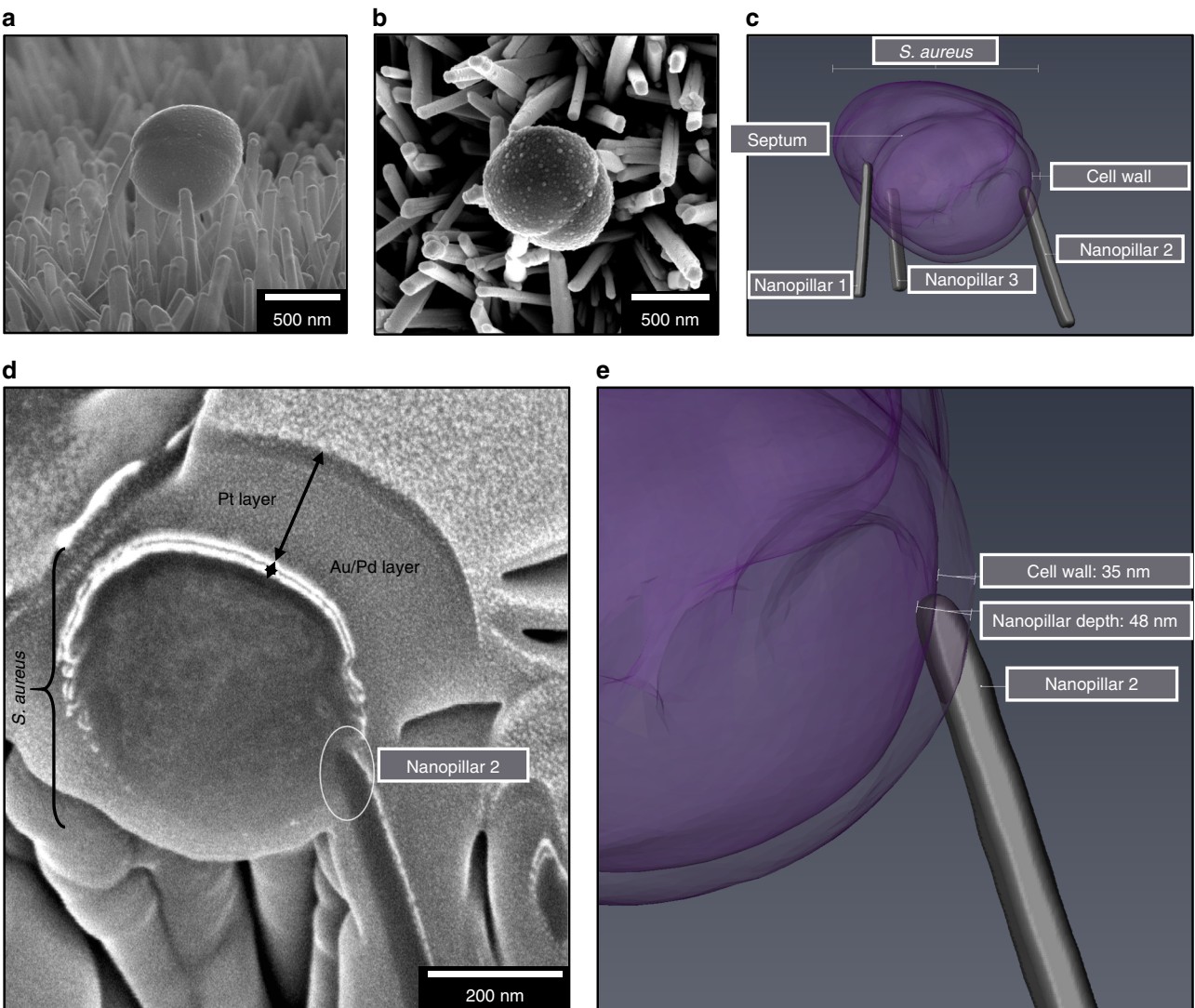

**Fig. 7 3D FIB-SEM reconstruction of *S. aureus*.** Automated FIB-SEM cross sectional analysis was performed on an *S. aureus* cell interacting with three nanopillars **a**, **b**. The focused ion beam produced 57 cross sections (30 nm each) that were imaged, and reconstructed in Avizo **c**. Analysis of *S. aureus* cross section #40 showed the tip of nanopillar 2 located ~50 nm into the cell, providing confirmation of cell wall and inner envelope penetration **d**, **e**. FIB-SEM analysis was performed on one *S. aureus* cell interacting with TiO$_2$ nanopillars.

and bacterial viability analysis (Fig. 8) confirmed that nanopillars had impaired bacterial growth. However, the relatively low frequency with which envelope deformation and penetration were observed could not account for the concomitant reductions in bacterial viability seen on nanowire arrays relative to controls. It was considered then that nanopillars might trigger a physiological response[19–21], reducing the capacity of bacteria to proliferate. To explore this hypothesis, TMT labelling and mass spectrometry analysis were performed to enable a direct quantitative comparison of *S. aureus* and *E. coli* proteomes after 24 h in the presence or absence of NW-850-5. Proteomic analysis revealed that 90 *S. aureus* proteins and 27 *E. coli* proteins exhibited differential expression with a $P \leq 0.05$, corresponding to 7 and 2% of the total proteomes, respectively (Supplementary Figs. 10a and 11a). Gene Ontology (GO) enrichment analysis was used to categorise each protein by molecular function, biological process and cellular component (Supplementary Figs. 10b and 11b). To uncover whether *S. aureus* and *E. coli* differentially expressed proteins (DEPs) were biologically connected, protein–protein interactions were mapped using the STRING application within Cytoscape (Fig. 9)[26,27]. In both instances, the number of interactions

(*S. aureus*, 421; *E. coli*, 14) was significantly higher than would be expected for a random set of proteins of similar size drawn from the genome. This verified that the *S. aureus* and *E. coli* DEP networks were biologically connected.

A number of the *S. aureus* DEPs are associated with protection from oxidative stress. Of note, superoxide dismutase, responsible for the conversion of superoxide anions into oxygen and hydrogen peroxide (H$_2$O$_2$)[28], had increased over two-fold in *S. aureus* recovered from NW-850-5. The abundance of other oxidative stress proteins had also increased in *S. aureus*, namely methionine sulfoxide reductase, responsible for repairing the oxidation of methionine residues[29], and bacterioferritin comigatory protein, a thiol peroxidase involved in the reduction of H$_2$O$_2$ (ref. [30]). Furthermore, the NADH-dependent flavin oxidoreductase (NWMN_0315) had increased nearly six-fold in the presence of NW-850-5. Expression of such proteins is induced by H$_2$O$_2$ stress[31]. SOS response proteins UvrA and UvrC, and bleomycin resistance protein had also increased significantly in the presence of NW-850-5, which mediate repair of DNA damage[32,33], along with histone-like DNA-binding protein HU, which is involved in DNA stabilisation under extreme environmental conditions[34]. In

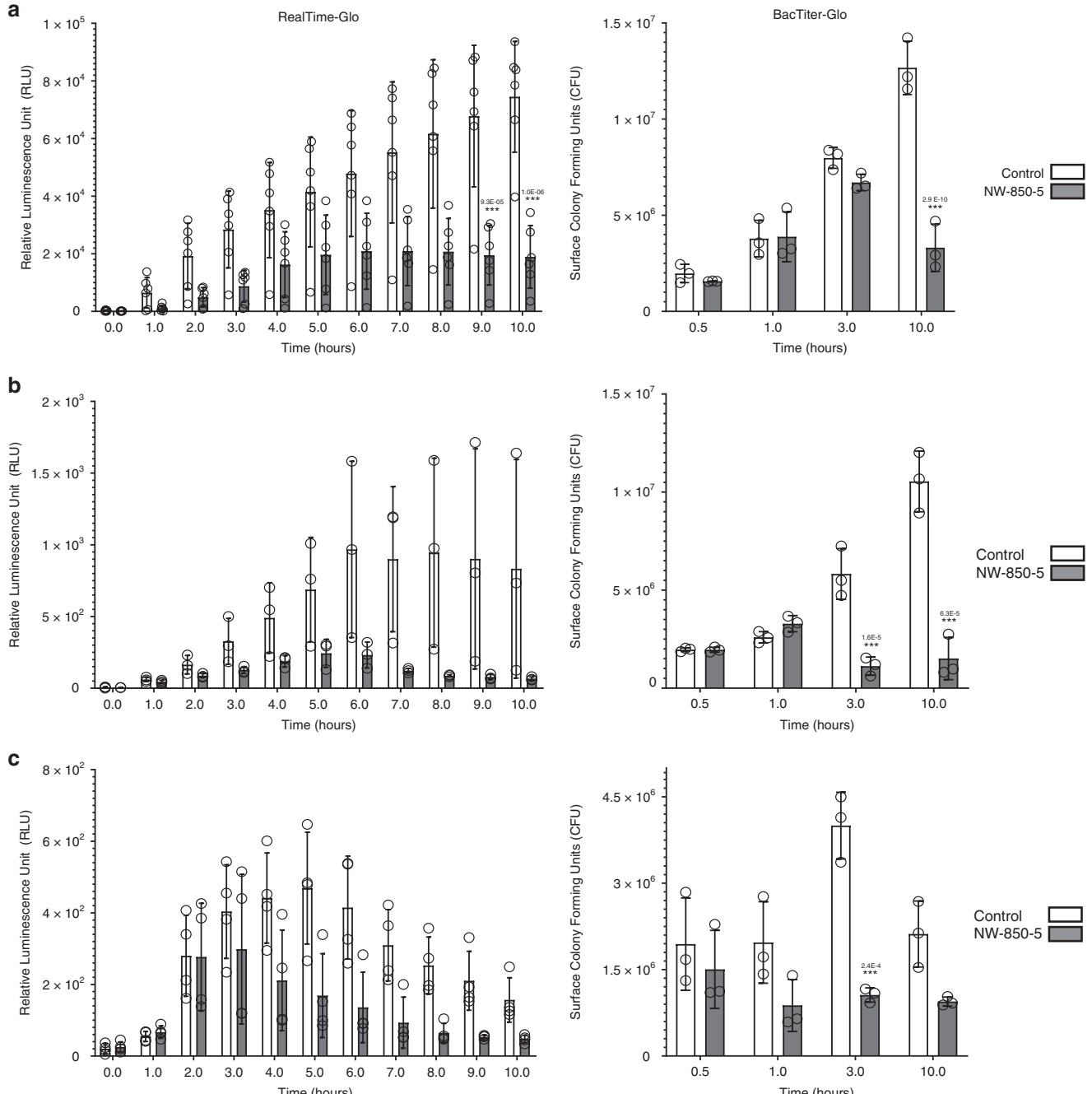

**Fig. 8 Determining bacterial viability on nanopillar surfaces.** Luminescence signals and CFU of *S. aureus* **a**, *E. coli* **b** and *K. pneumoniae* **c** incubated on control or nanopillar TiO$_2$ surfaces for up to 10 h, as determined by real-time or endpoint assays, respectively. Values are given as mean ± standard deviation and individual data points for each mean are shown. *** indicates $P \leq 0.001$ relative to control, as determined by one-way ANOVA and Tukey HSD post hoc test. BacTiter-Glo experiments $n = 3$; RealTime-Glo experiments $n = 6$ (*S. aureus*), $n = 3$ (*E. coli*) and $n = 4$ (*K. pneumoniae*). Total surface CFU were determined from disc areas of 0.64 cm$^2$. Exact *P*-values are indicated for each significant time point.

addition to oxidative stress, another key theme of *S. aureus* DEPs was protein expression. A significant proportion (33%) of DEPs were involved in protein translation, secretion and translocation, all of which had been upregulated in the presence of NW-850-5.

As for *S. aureus*, oxidative stress was a key theme of the DEPs identified for *E. coli*. Subunit A (ClpA) of the ATP-dependent protease (ClpAP), which mediates unfolding and transloca-tion of abnormal proteins[35], had decreased significantly in the presence of NW-850-5. Oxidative stress is known to inactivate such chaperone proteins[36]. Concomitantly, chaperedoxin had

increased in the presence of NW-850-5, which is known to protect against the irreversible protein aggregation and oxida-tion[36]. Another notable *E. coli* DEP was surface composition regulator GlgS. This is a negative regulator of type-1 fimbriae and flagella production[37], and was increased significantly in *E. coli* recovered from NW-850-5. It is possible that the increased abundance of GlgS could account for the absence of extracellular appendages observed for *E. coli* on NW-850-5 under SEM. Nanoscale variations in surface topography are reported to significantly alter protein expression in bacteria, and Rizzello and colleagues found that *E. coli* adhered to gold nanoparticle

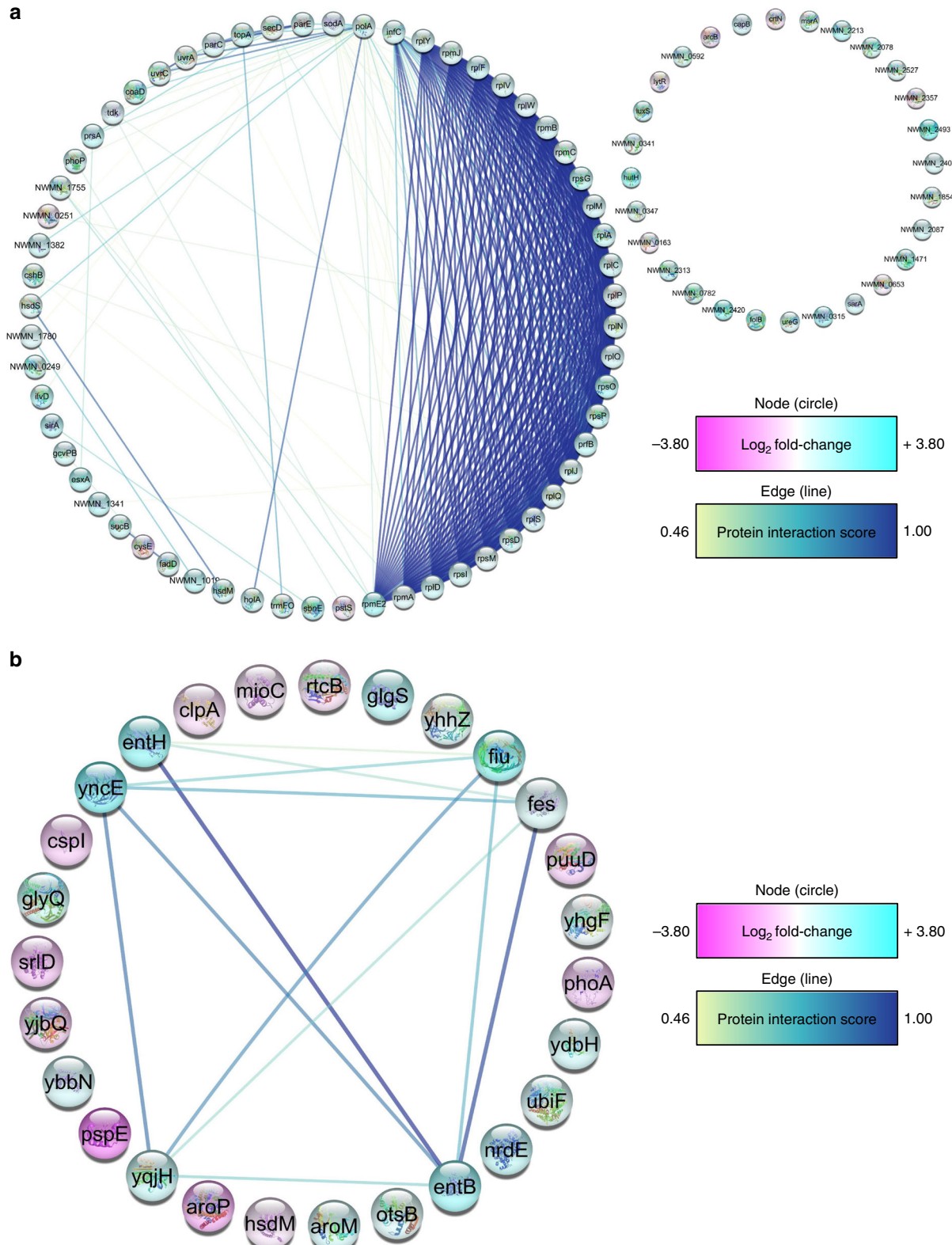

**Fig. 9 S. aureus and E. coli proteomic response to nanopillar surface NW-850-5.** The predicated functional partners among *S. aureus* **a** and *E. coli* **b** DEPs are shown, where individual proteins are represented in circles (nodes), and protein–protein interactions (PPI) are represented by the connecting lines (edges). DEPs have been coloured according to the log2 fold change in protein abundance (pink = decreased abundance and blue = increased abundance), and node colour indicates the confidence of supporting interaction evidence (0.15 = low confidence, 0.40 = medium confidence, 0.70 = high confidence and 0.9 = highest confidence). For *S. aureus*, 60 DEPs displayed at least one interaction at a confidence level between 0.46 and 1.00, and the PPI enrichment *P*-value was 1.0e−16. For *E. coli*, six proteins had at least one interaction at a medium to high confidence and had a PPI enrichment *P*-value of 9.54e−06. PPI enrichment *P*-values were determined using a hypergeometric test and corrected for multiple testing using the method of Benjamini and Hochber[54]. Data are representative of one experimental replicate, performed in triplicate.

surfaces expressed significantly lower levels of type-1 fimbriae. Concomitantly, proteins involved in DNA and membrane stress were upregulated[19].

**Determination of oxidative stress response to NW-850-5**. Taken together, proteomic analysis, while preliminary, suggested that TiO$_2$ nanopillars had induced an oxidative stress response within *S. aureus* and *E. coli*. A possible explanation for oxidative stress could be increased reactive oxygen species (ROS), as has been reported on CNT surfaces[38]. To investigate this as a possible mechanism for reduced viability, the ROS-Glo assay was used to measure the levels of H$_2$O$_2$ (an oxidative stress marker) produced from *S. aureus* and *E. coli*, following 24-h incubation on NW-850-5 or control discs. The levels of H$_2$O$_2$ in *S. aureus* and *E. coli* culture media were significantly higher on NW-850-5 compared to control discs, corresponding to 1.6- and 3.8-fold increases, respectively (Fig. 10a). Given that ROS mediate damage of DNA, lipids and proteins[32,39], the increased quantity of H$_2$O$_2$ could be expected to induce changes in bacterial envelope morphology. Following 24-h incubation, the density and coverage of *S. aureus* and *E. coli* were notably higher on control compared to NW-850-5 discs (Fig. 10). At higher magnification, the characteristic morphologies of *S. aureus* (coccoid) and *E. coli* (bacillus) were observed on control surfaces (Fig. 10b, d), with evidence of microcolony and biofilm formation. In contrast, the morphology of *S. aureus* and *E. coli* on NW-850-5 discs was less rigid and did not conform to the defined shapes observed on control discs (Fig. 10c, e). This was particularly evident for *E. coli*, which appeared to be sinking into the nanopillars. These morphologies were consistent with previous studies that have proposed ROS-mediated cell death on CNTs (ref. [38]).

## Discussion

Elucidating the mechanisms that govern bacterial cell death on nanotextured materials is essential for improving antibacterial performance. This, in turn, will guide the design of next-generation biomaterials that effectively inhibit biofilm growth, without reliance on antibiotics. To date, natural and synthetic nanotextured materials are reported to mediate bactericidal activity via mechanical rupture of the bacterial cell, resulting in lysis and cell death[1–6]. In this study, analysis of bacterial cross sections using TEM and tomography identified clear signs of localised nanopillar-induced envelope deformation and penetration. However, there was no evidence to support that bacterial lysis had occurred, even with the concomitant reductions in bacterial viability. Similar cases of nanopillar-induced envelope deformation and penetration were identified by FIB cross sectional analysis. These observations were most prominent in Gram-negative bacteria *E. coli* and *K. pneumoniae*, which possess thin peptidoglycan layers (≈5 nm). Nonetheless, there was strong evidence that nanopillars were also able to penetrate *S. aureus*, albeit at a much lower frequency. The reduced susceptibility of *S. aureus* to nanopillar deformation may partly be explained by the increased peptidoglycan thickness, providing increased rigidity[6] and higher turgor pressure; indeed, one study reported a lower total creep deformation in *B. subtilis* compared to *E. coli*[40]. In another study, the force required to rupture *Staphylococcus epidermidis* was nearly four-fold greater than for *E. coli*, measured at 13.8 μN and 3.6 μN, respectively[41]. Although *S. aureus* appeared less susceptible to nanopillar-induced killing than *E. coli* or *K. pneumoniae*, further investigations are required to assess the antibacterial properties of nanopillar surfaces over longer incubations and to establish their performance against microorganisms with thicker cell walls (e.g., fungi and spores).

Although envelope deformation and penetration were confirmed in this study, the frequency of such events was low and did not result in mechanical rupture and cell lysis. Therefore, these interactions could not solely account for the reductions in cell viability that were recorded. However, proteomic analyses identified a number of *S. aureus* and *E. coli* DEPs associated with oxidative stress. This was further supported by ROS-Glo analysis, which revealed that H$_2$O$_2$ levels were significantly higher in *S. aureus* and *E. coli* incubated on NW-850-5 than on controls. Bacterial ROS production is frequently observed following exposure to bactericidal antibiotics[42–45]. This mechanism is reported to be self-amplifying, meaning that ROS production is sustained, even when the initial stressor is removed[39]. Bacterial oxidative stress may also be activated by other stressors, such as copper[46] or the accumulation of misfolded proteins[28]. Given that bacterial oxidative stress has overlapping activating signals[47], it is possible that the increased abundance of oxidative stress proteins and elevated levels of H$_2$O$_2$ reported here are directly attributable to the effects of the TiO$_2$ nanopillars.

Previous research has established a connection between ROS-mediated bacterial cell death and nanostructure contact. In one study, CNTs were shown to increase ROS production in *Candida albicans*, *P. aeruginosa* and *S. aureus* following 24-h incubation. ROS production was proposed to have been induced by physical contact with the CNTs, which were reported to wrap around bacteria rather than pierce the envelope. Of note, ROS production was shown to occur independently of photocatalytic activation[38]. No photocatalytic activity was detected on the TiO$_2$ nanopillar surfaces in this study, as the luminescence generated by the ROS-Glo assay was negligible in the absence of bacteria. This is most likely attributable to the rutile crystal structure of TiO$_2$ nanopillars, which generally possesses lower photocatalytic activity than anatase polymorphs[48,49].

This study has highlighted several key findings that have important implications for the development of antimicrobial nanotopographies for biomedical applications. Firstly, the mechanistic basis of contact killing is multifactorial (Supplementary Fig. 12) and nanotopography dependent. While deformation and subsequent penetration of the bacterial envelope by nanopillars was confirmed in this study, these mechanisms did not result in mechanical rupture or cell lysis. Another notable mechanism identified in this study was nanopillar-induced cell impedance, which is expected to reduce the capacity of bacteria to replicate on nanopillar surfaces, and thus could enhance the anti-biofilm properties of nanopillar surfaces. Furthermore, our analyses provide compelling evidence that nanopillars can induce oxidative stress within bacterial cells upon contact, the cumulative effects of which could impair processes, such as bacterial growth and biofilm formation. Indeed, this could account for the time-dependent reductions in bacterial viability. Better understanding of this mechanism could prove invaluable for improving the antibacterial performance of nanotextured materials. Thus, further research is needed to examine more closely the link between ROS generation, nanotopography design and bactericidal activity. It is also evident from this study that multiple experimental approaches should be exploited to robustly assess all aspects of bacterial physiology, so as to obtain a comprehensive assessment of the antibacterial properties of nanotextured materials. This will reduce experimental bias toward a single mechanism of action, and provide a more complete and accurate understanding of the antimicrobial mechanism of nanotopographies.

## Methods

**Generation of TiO$_2$ nanopillars on titanium by thermal oxidation**. Grade 5 titanium alloy (Ti-6Al-4V, Titanium Metals Ltd) samples (0.64 cm$^2$) were machine polished (Struers® TegraForce-1) using decreasing silicon carbide grit sizes (#80,

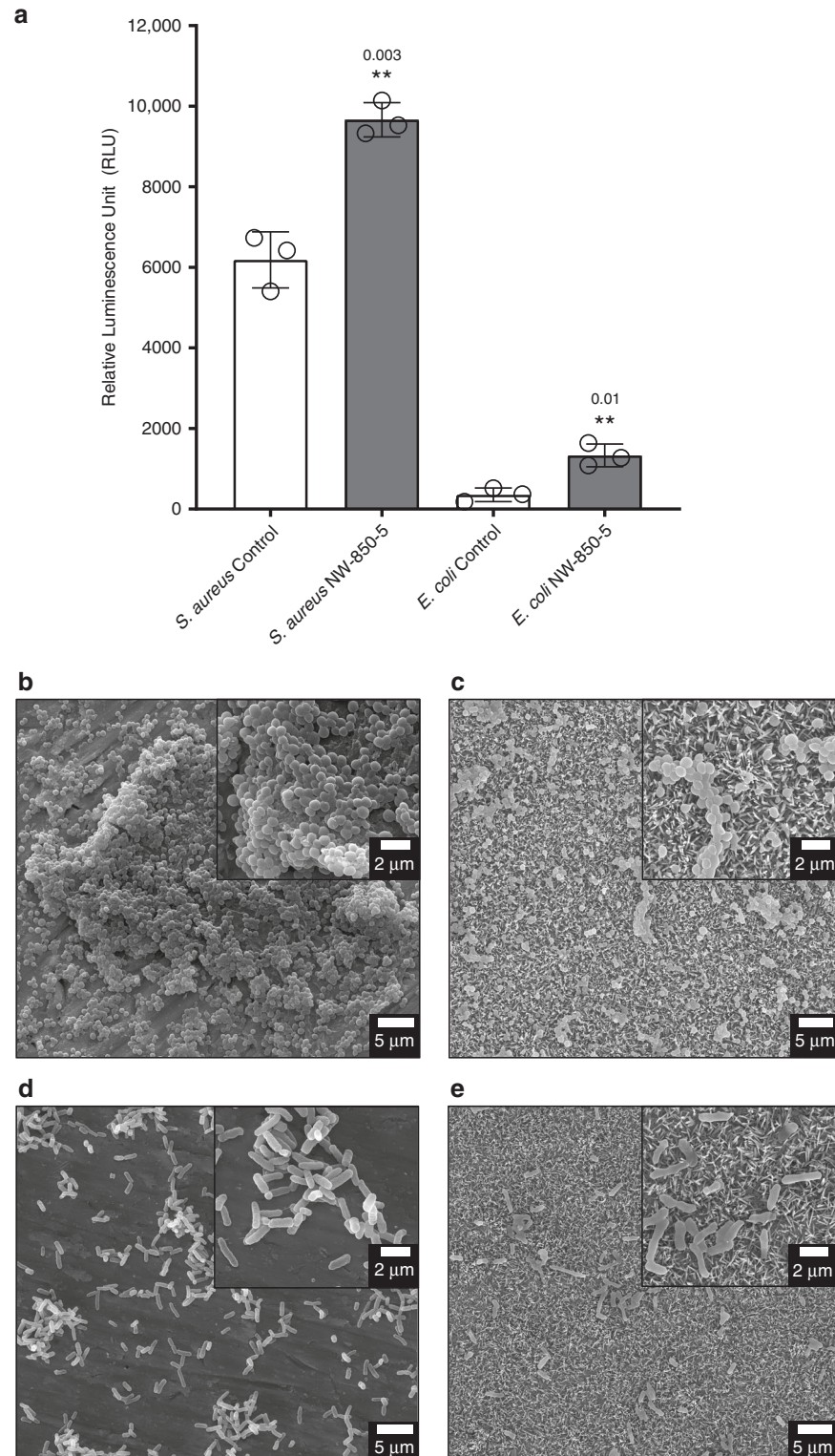

**Fig. 10 Determination of ROS production in response to nanopillar surfaces.** *S. aureus* or *E. coli* were incubated for 24 h on control or NW-850-5 surfaces under the same growth conditions used for proteomic analysis. Following 24-h incubation, the levels of $H_2O_2$ were determined by ROS-Glo **a**. Values given are mean ± standard deviation and individual data points for each mean are shown. ** indicates $P \leq 0.01$ relative to control for *S. aureus* ($P = 0.003$) and *E. coli* ($P = 0.01$), as determined by a two-sided Student's *t*-test; $n = 3$ performed in duplicate. Representative SEM micrographs of *S. aureus* **b**, **c** and *E. coli* **d**, **e** incubated on control or NW-850-5 are shown. Insets show higher magnification SEM images of each surface.

#500, #1200, #2000 and #4000). To remove surface contaminants, titanium discs were placed inside a digital ultrasonic bath (Grant Scientific XUB series) in distilled water (dH$_2$O), and the samples were cleaned at 40 °C for 15 min using 100% power. Following ultrasonication, titanium samples were placed in ethanol (analytical reagent grade (99.99%), Fisher Scientific) for 10 min before air drying. Polished and

cleaned titanium samples were sealed inside a horizontal alumina tube (120 cm × 11 cm outer × 9 cm inner) positioned in a furnace (Elite Thermal Systems Ltd). Prior to thermal oxidation, the furnace was purged with inert argon gas (Ar) to remove oxygen and achieve a one-directional flow. Following purging, heating was initiated at 15 °C per minute until a pre-defined maximum was reached; in this

study, a temperature of 850 °C was used. Once the final temperature was reached, Ar was redirected into a sealed Duran$^{TM}$ vessel containing liquid acetone (analytical reagent grade (99.99%), Fisher Scientific), maintained at 25 °C. This generated an acetone vapour phase, which initiates the oxidation reaction with titanium samples. Following completion of the heating programme, furnace cooling began. The flow of Ar was maintained at a constant rate until room temperature was reached.

**Nanopillar characterisation**. A Quanta 200 field emission gun (FEG) SEM (FEI) was used to determine the dimensions of nanopillars produced by thermal oxidation. To establish the homogeneity of nanopillar coverage and the reproducibility between thermal oxidation batches, top view electron micrographs were taken of discs from three independent batches; micrographs were taken at three locations per disc. FIJI software was used to quantify the density of nanopillars per μm$^2$, using the find maxima function. Tilted (40°) electron micrographs were used to estimate the length and tip diameter of nanopillars, with three locations captured per disc, from three independent batches. A total of 20 length and 20 diameter measurements were recorded from each micrograph. These data were averaged from each batch to estimate the mean nanopillar length and diameter. The elemental composition of nanopillars was determined using EDX in line-scanning mode on a JEOL JEM-2100F FEG TEM. The crystal structure of TiO$_2$ nanopillars was analysed with grazing-incident X-ray diffraction. Titanium samples were mounted 0.243 m from the detector and tilted at 0.7°. An X-ray beam (wavelength of 885.6 nm) was shot at the sample and the diffracted X-rays were detected and imaged at two positions, 8° and 30°. The images were converted into 1D profiles by azimuthal integration using a pyFAI package. The ratio of anatase to rutile was determined using the Spurr and Myers equation[50], by calculating the area underneath characteristic anatase ($2\theta = 26.5$) and rutile ($2\theta = 27.5$) peaks.

**Bacterial strains and culture conditions**. *S. aureus* strain Newman, *S. aureus* SH1000 (ref. [51]), *E. coli* strain K12, *E. coli* TOP10 (ref. [52]) and *K. pneumoniae* (clinical isolate kindly provided by M. Avison) were used in this study. Broth cultures were incubated for 16 h at 37 °C, 220 rpm, subcultured to OD$_{600}$ 0.1, and grown to mid-exponential phase. Bacteria were routinely cultured in Luria Bertani broth (BD Biosciences) apart from for the BacTiter-Glo assay, for which Mueller Hinton broth (Sigma-Aldrich) was used. Titanium samples were sterilised in absolute ethanol, washed in dH$_2$O and dried prior to inoculation with bacterial suspensions. For all bacterial viability assays, titanium discs were incubated statically at 37 °C.

**Determination of bacterial viability**. For BacTiter-Glo analysis, sterile titanium discs were placed inside white 24-well plates (PerkinElmer, MA, USA) and inoculated with 25 μL of bacterial culture (equivalent to 10$^6$–10$^7$ CFU), forming a meniscus. Titanium discs were incubated for 0.5, 1, 3 and 10-h time points at 37 °C under static conditions inside a humidity chamber. Following incubation, equilibrated BacTiter-Glo reagent was added in a 1:1 ratio to the bacterial suspension and incubated for 5 min in the dark, prior to RLU measurements. Quantification of the total number of bacterial cells on each titanium disc was based on standard curves relating RLU to CFU (Supplementary Fig. 9). For RealTime-Glo assays, sterile titanium discs were inoculated with 50 μL of bacterial culture (equivalent to 10$^6$–10$^7$ CFU), supplemented with 1× MT Cell Viability Substrate and 1× NanoLuc® Enzyme. Each 24-well plate was sealed and luminescence measurements were recorded continuously for 10 h at 37 °C under static growth conditions. For both assays, luminescence was recorded using an automated Infinite® 200 PRO microplate reader (Tecan, Männedorf, Switzerland). Luminescence was recorded using automatic attenuation, 1 s integration time and 0.15 s settle time.

**SEM sample preparation and imaging**. Bacteria on titanium discs were fixed at 4 °C in 2.5% EM grade glutaraldehyde (Agar Scientific Ltd. Essex, UK) buffered with 0.1 M sodium cacodylate (Acros Organics, New Jersey, USA). Following overnight fixation, samples were washed (3 × 5 min) in 0.1 M sodium cacodylate buffer and dehydrated in a graded ethanol series of 25%, 50%, 70%, 90 and 100% (Sigma-Aldrich, St. Louis, MO, USA). Samples were then critically point dried using a Leica CPD300, following an established protocol for microbial cells[53]. Prior to image acquisition on a Quanta 200 FEG-SEM, titanium discs were mounted to 0.5″ aluminium stubs using colloidal silver paste (Agar Scientific Ltd. Essex, UK), before being coated with a 10 nm chromium layer using an Emitech K757X sputter coater system.

**Sample preparation for TEM and FIB-SEM**. Following overnight fixation in 2.5% EM grade glutaraldehyde at 4 °C, samples were washed (3 × 5 min) in 0.1 M sodium cacodylate buffer prior to OTO (osmium tetraoxide–thiocarbohydrazide–osmium) processing. Briefly, this method included post fixation in equal volumes of 4% osmium tetraoxide (Agar Scientific Ltd. Essex, UK) and 3% potassium ferrocyanide (Sigma-Aldrich, St. Louis, MO, USA) for 60 min on ice. Following post fixation, samples were rinsed (3 × 5 min) in dH$_2$O before incubating with thiocarbohydrazide (Sigma-Aldrich, St. Louis, MO, USA) for 20 min. Additional dH$_2$O washing steps (3 × 5 min) were applied before incubation in 2% aqueous osmium for 30 min at room temperature. Following

OTO processing, bacterial samples were stained in 1% aqueous uranyl acetate (1 h at 4 °C) followed by lead aspartate (200 μM) for 30 min in the dark. Between these steps, washing with dH$_2$O was performed. After the final washing step, bacterial samples were dehydrated and dried as previously outlined.

**TEM sectioning and imaging**. Bacteria incubated on control titanium discs were recovered and pelleted prior to staining and microtomy, whereas bacteria incubated on TiO$_2$ nanopillar surfaces were stained, embedded and sectioned, while attached to the titanium disc. All samples to be analysed using TEM were embedded in epon 812 resin (TAAB Laboratories Equipment Ltd. Berkshire, UK) for 90 min on a TAAB rotator type N, held at 2 rpm. Samples were then placed at 60 °C for 48 h to allow resin polymerisation. Resin embedded samples were sectioned using an Ultra Diamond Knife (DiATOME) on a Leica EM UC6 ultra-microtome. These were transferred to pioloform-coated TEM grids for microscopy. To analyse the ultrastructure of bacterial cells adhered to TiO$_2$ nanopillar surfaces, samples were loaded into a Tecnai 12 FEI 120 kV BioTwin Spirit TEM. An electron beam with an accelerating voltage of 120 kV was used to capture images. For tilt series acquisition, TEM samples were transferred to a Fischione tomography holder and loaded into a Tecnai 20-FEI 200 kV Twin Lens scanning TEM. Tilt series projections were acquired in bright-field mode. Dedicated FEI tomography software was used to capture the tilt series data. An electron beam with an accelerating voltage of 200 kV was used to capture images.

**FIB sectioning and imaging**. Titanium samples were loaded into the vacuum chamber of a Strata FIB201 (FEI). When searching for bacteria of interest, an electron beam with an accelerating voltage of 5 kV and current of 98 pA was used. To preserve bacterial morphology, electron and ion-assisted platinum deposition was performed at 27.5 pA, creating a protective layer 0.5 μm thick. Once coated, rough cut trenches were milled around the bacteria to depths of 250 nm using an accelerating voltage of 5 kV, and current of 6.5 nA. Auto Slice and View software (FEI) was used to carry out sequential sectioning of *S. aureus* and *E. coli* in 30 nm slices. This was performed with an accelerating voltage of 5 kV, and beam current of 47.5 pA. Images of each section were acquired using electron beam accelerating voltages of 5 kV and current of 98 pA. Secondary electron emission was detected using an Everhart-Thornley detector.

**TEM and FIB-SEM image processing and 3D volume reconstruction**. TEM cross sectional analysis was used to determine the frequency of nanopillar-induced envelope deformation and penetration in *S. aureus*, *E. coli* or *K. pneumoniae*. Frequencies are reported as percentages of the total cells analysed; 59 (*S. aureus*), 42 (*E. coli*) or 80 (*K. pneumoniae*). To construct tomograms of bacteria, raw tilt series data sets were processed using eTomo software. Tomograms were uploaded into Avizo v9.7.0 (FEI), and the segmentation editor was used to generate 3D volumes of each tomogram. The slice and view data acquired from sequential FIB milling was processed using the FIB-stack wizard tool in Avizo v9.7.0 (FEI). Briefly, this tool facilitates in aligning the FIB-stack and correcting geometrical artefacts, such as stage tilt foreshortening and/or vertical shift. Following data alignment and shearing correction within the FIB-stack wizard, image denoising was performed using a non-local means filter. Within Avizo segmentation editor, regions of bacteria and nanopillar were assigned as different materials, and were manually traced in consecutive orthoslices using the brush tool. The interpolation function was not used to join neighbouring orthoslices. The generate surface function was used in repeatable algorithm mode with smoothing deselected to produce 3D volumes of bacteria and nanopillars. 3D volumes were then simplified using the simplification editor in default mode. The smooth surface function was then applied to simplified 3D meshes of bacteria and nanopillars, with iteration and lambda functions set to 50–200 and 0.6, respectively.

**Protein extraction from *S. aureus* and *E. coli***. Titanium discs (25 cm$^2$) were placed inside sterile petri dishes (94 mm by 16 mm) and were inoculated with 1 mL *S. aureus* or *E. coli* suspensions containing 10$^7$–10$^8$ cells. Titanium discs were incubated for 24 h at 37 °C under static conditions inside a humidity chamber. Following incubation, titanium samples were rinsed in 20 mL Tris-HCl buffer to recover bacteria. Cells were harvested by centrifugation (5000 rpm, 10 min) and resuspended in Tris-HCl. To extract *S. aureus* proteins, Tris-HCl (200 μL) was supplemented with 200 μg/mL lysostaphin (Sigma-Aldrich, St. Louis, MO, USA), and suspensions were incubated for 60 min at 37 °C before cellular debris was removed by centrifugation at 13,300 rpm for 20 min at 4 °C. For *E. coli* proteins, cell suspensions (500 μL) were snap frozen in liquid nitrogen and transferred to a pre-cooled 5 mL Teflon flask containing one tungsten carbide bead. Homogenisation of frozen bacteria was performed at 2000 rpm for 2.5 min, using a Micro-Dismembrator (Sartorius, Göttingen, Germany). This resultant powder was resuspended in 1 mL Tris-HCl (pH 7.2) and cellular debris was removed by centrifugation at 13,300 rpm for 20 min at 4 °C. Soluble *S. aureus* and *E. coli* protein fractions were then transferred to an Eppendorf tube and stored at −80 °C.

**TMT labelling and high pH reversed-phase chromatography**. An equal volume of each protein sample was digested overnight at 37 °C with 2.5 μg trypsin, labelled with TMT ten plex reagents according to the manufacturer's protocol (Thermo

Fisher Scientific), and the labelled samples pooled. An aliquot of the pooled sample was evaporated to dryness, resuspended in 5% formic acid and then desalted using a SepPak cartridge according to the manufacturer's instructions (Waters, USA). Eluate from the SepPak cartridge was again evaporated to dryness and resuspended in buffer A (20 mM ammonium hydroxide, pH 10) prior to fractionation by high pH reversed-phase chromatography using an Ultimate 3000 liquid chromatography system (Thermo Scientific). In brief, the sample was loaded onto an XBridge BEH C18 Column (130 Å, 3.5 µm, 2.1 mm × 150 mm, Waters, UK) in buffer A and peptides eluted with an increasing gradient of buffer B (20 mM ammonium hydroxide in acetonitrile, pH 10) from 0–95% over 60 min. The resulting fractions were evaporated to dryness and resuspended in 1% formic acid prior to analysis by nano-LC MSMS using an Orbitrap Fusion Tribrid mass spectrometer (Thermo Scientific).

**Nano-LC mass spectrometry**. High pH reversed-phase fractions were further fractionated using an Ultimate 3000 nano-LC system in line with an Orbitrap Fusion Tribrid mass spectrometer (Thermo Scientific). In brief, peptides in 1% (v/v) formic acid were injected onto an Acclaim PepMap C18 nano-trap column (Thermo Scientific). After washing with 0.5% (v/v) acetonitrile/0.1% (v/v) formic acid, peptides were resolved on a 250 mm × 75 µm Acclaim PepMap C18 reverse-phase analytical column (Thermo Scientific) over a 150 min organic gradient, using seven gradient segments (1–6% solvent B over 1 min, 6–15% B over 58 min, 15–32% B over 58 min, 32–40% B over 5 min, 40–90% B over 1 min, held at 90% B for 6 min and then reduced to 1% B over 1 min) with a flow rate of 300 nl/min. Solvent A was 0.1% formic acid and Solvent B was aqueous 80% acetonitrile in 0.1% formic acid. Peptides were ionised by nano-electrospray ionisation at 2.0 kV using a stainless steel emitter with an internal diameter of 30 µm (Thermo Scientific) and a capillary temperature of 275 °C.

All spectra were acquired using an Orbitrap Fusion Tribrid mass spectrometer controlled by Xcalibur 2.0 software (Thermo Scientific) and operated in data-dependent acquisition mode using an SPS-MS3 workflow. FTMS1 spectra were collected at a resolution of 120,000, with an automatic gain control (AGC) target of 200,000 and a max injection time of 50 ms. Precursors were filtered with an intensity threshold of 5000, according to charge state (to include charge states 2–7) and with monoisotopic precursor selection. Previously interrogated precursors were excluded using a dynamic window (60 s +/− 10 ppm). The MS2 precursors were isolated with a quadrupole mass filter set to a width of 1.2 $m/z$. ITMS2 spectra were collected with an AGC target of 10,000, max injection time of 70 ms and CID collision energy of 35%. For FTMS3 analysis, the Orbitrap was operated at 50,000 resolution with an AGC target of 50,000 and a max injection time of 105 ms. Precursors were fragmented by high energy collision dissociation at a normalised collision energy of 60% to ensure maximal TMT reporter ion yield. Synchronous Precursor Selection (SPS) was enabled to include up to five MS2 fragment ions in the FTMS3 scan.

**Proteomic data analysis**. The raw data files were processed and quantified using Proteome Discoverer software v2.1 (Thermo Scientific) and searched against the UniProt S. aureus (strain Newman) database (downloaded October 2018; 2584 entries) or the Uniprot E. coli (strain K12) database (downloaded February 2019; 4469 entries), using the SEQUEST algorithm. Peptide precursor mass tolerance was set at 10 ppm, and MS/MS tolerance was set at 0.6 Da. Search criteria included oxidation of methionine (+15.9949 Da) as a variable modification and carbamidomethylation of cysteine (+57.0214 Da) and the addition of the TMT mass tag (+229.163 Da) to peptide N-termini and lysine as fixed modifications. Searches were performed with full tryptic digestion and a maximum of two missed cleavages were allowed. The reverse database search option was enabled, and all data were filtered to satisfy a false discovery rate of 5%. Based on the protein abundance values from three biological replicates of each sample, a Student's t-test was used to determine the significance of protein abundance fold changes between S. aureus and E. coli incubated on NW-850-5 and control surface. Proteins with $P \leq 0.05$ were deemed significant. Blast2Go software was used to categorise S. aureus and E. coli DEPs based on GO terms; DEPs were grouped by Level 2 GO. Protein–protein interactions were investigated using the functional protein association network tool (STRING v11) within Cytoscape.

**ROS-Glo™ H₂O₂ assay**. As for the proteomic studies, S. aureus and E. coli were incubated on titanium discs (NW-850-5 or control) for 24 h. After 18 h, the $H_2O_2$ substrate solution was added to the discs, as per the manufacturer's protocol, before placing back into the incubator at 37 °C for the remaining 6 h. During this time, the $H_2O_2$ substrate reacts with $H_2O_2$, forming a luciferin precursor. After 24 h, the ROS-Glo detection reagent was added. This converts the luciferin precursor into luciferin, which is then used by the Ultra-Glo™ Recombinant Luciferase to generate a luminescent signal that is proportional to the quantity of $H_2O_2$. Luminescence measurements were recorded using an automated Infinite® 200 PRO microplate reader. Luminescence was recorded using automatic attenuation, 1 s integration time and 0.15 s settle time.

**Statistical analysis**. Statistical analyses were performed using IBM SPSS statistical package (version 25). For bacterial viability testing, statistically significant differences were determined using one-way analysis of variance (ANOVA), with a Tukey HSD post hoc test. For proteomic and oxidative stress response analyses, statistically significant differences were determined by Student's t-test. Statistically significant differences were attributed to variables with $P \leq 0.05$. Unless otherwise stated, values given are mean ± standard deviation and are representative of three independent experimental replicates ($n = 3$), performed in triplicate.

**Reporting summary**. Further information on research design is available in the Nature Research Reporting Summary linked to this article.

## Data availability
Source data underlying Figs. 8 and 10, and Supplementary Fig. 9, are available as a Source Data file. The mass spectrometry proteomics data have been deposited to the ProteomeXchange Consortium via the PRIDE partner repository with dataset identifier PXD017078. The UniProt S. aureus (strain Newman) database (downloaded October 2018; 2584 entries, Proteome ID: UP000006386) and the Uniprot E. coli (strain K12) database (downloaded February 2019; 4469 entries, Proteome ID: UP000000625) were used in this study. Other data supporting the findings of this study are available from the corresponding authors upon request.

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

## Acknowledgements

We acknowledge funding from the Medical Research Council (MRC) Doctoral Training Programme (J.J.). BS and AHN would like to thank the MRC for funding (MR/N010345/1 & MR/S010343/1). We thank Wolfson Bio-imaging Facility and Proteomics Facility at the University of Bristol for their help with electron microscopy and quantitative proteomic analysis respectively. We thank the Henry Royce Institue for FIB-SEM access funding.

## Author contributions

B.S. conceived the project and contributed to the editing of the manuscript. A.H.N. conceived the project and contributed to the editing of the manuscript. P.V. conceived the project. J.J. wrote the manuscript, performed all the experiments and data analysis, except for tilt series collection, cross section generation, and slice and view data collection, which was performed by J.M., C.N. and A.G., respectively.

## Competing interests

The authors declare no competing interests.

## Additional information

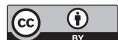

