## [Peer Review File · Nature Communications]

Reviewers' comments:

Reviewer #1 (Remarks to the Author):

The manuscript by Su et al. addresses the very important problem of finding basic solutions for preventing hospital-acquired infections through novel antibacterial implant surfaces. The new nanowire materials surfaces presented here are bioinspired by cicada and dragonflies nanoarrays. The authors show convincingly how nanowire arrays induce deformation and penetration of Gram-positive and Gram-negative envelopes but do not rupture or lyse the microbes. In addition it is shown by proteomic analysis and cell test that the titanium nanowires also have capacity to impede bacterial division and trigger oxidative stress responses. The discovery of the latter mechanism can be considered groundbreaking.

The paper is very well written and the claims are supported very well by experimental evidence and a clear discussion. The experiments are very well designed and well conducted. The TEM and SEM results are very impressive. The study delivers new insights into mechanisms previously unknown this may have potential to have major impact for different fields. Therefore, the presented results are highly useful for others in the community and the wider fields of materials science and microbiology and medicine as a whole.

I fully support publication of the manuscript in Nature Communications after some minor revisions described in the following:

- The authors present in Supplementary Figure 2 a GXID spectrum and state that 98% of the nanowires are rutile. Yet, the other signals in the diffraction pattern are (for example alpha titanium) are very strong. This seems like a contradiction. How can this be explained?
- What is the nature of the dark bands that can be seen in the nanowires in Fig. 5a? Do they indicate composition fluctuations?
- How do the presented results compare to related previous results from Dewald et al. in *Colloids and Surfaces B: Biointerfaces* 163, 201-208 and Lüdecke et al. 145 (2016) 617-625? These use related materials and therefore these related works should at least be mentioned.
- Can the author elaborate a bit more for which microbes the presented approach will be effective and also what its limits are (for example microbes that will not be effected).
- Why are pili and fimbria absent for the microbes?

Reviewer #2 (Remarks to the Author):

The submitted article investigates the bacterial responses to titanium nanowire arrays using a range of techniques including electron tomography, biochemical assays and quantitative proteomic analysis. The authors determined the morphological and physiological responses of bacteria to dragonfly mimetic nanowire arrays, constructed 3D visualisations of bacteria adhered to nanowire arrays, determined changes in protein expression and quantified the growth and viability of *S. aureus*, *Staphylococcus epidermidis*, *E. coli* and *Klebsiella pneumoniae* populations.

The major claims of the report are as follows;

1) The nanowire arrays induce deformation and penetration of Gram-positive and Gram-negative envelopes, but do not rupture or lyse bacteria. This is a really interesting point and it is the first time that this has been demonstrated clearly. Many authors make suggestions that biomimetic or nanopillars surfaces are causing cell rupture, but do not prove it. This paper clearly does, and this is novel and will be of great interest to others in this research area and the wider research community.

2) Nanowires also have capacity to impede bacterial division and trigger oxidative stress responses. This was clearly proven. Why was this assay selected rather than others? It needs

defending why only ROS was evaluated.

However, although this is an excellent manuscript, that is novel, truly multidisciplinary, well written and well thought out, there are a few points to be addressed.

The proteomics data is important and the results from this part of the work are missed in the abstract, and throughout much of the paper. These results need weaving throughout the paper, especially in terms in what they contribute to the understanding of the bacterial processes.

Otherwise this looks like a separate piece of work.

The results begin with *S. epidermidis* included, then they disappear, only to be picked up later on in the manuscript. Throughout the paper, either refer to all the bacterial results or remove the bacteria that there is only partial data for.

With regards the colony forming units (CFU), it would be advantageous to know what area coverage this relates too.

The statistical analysis is included and satisfactory.

As such the manuscript should be highly considered for publication following minor corrections.

Reviewer #3 (Remarks to the Author):

This manuscript addresses the important topic of how bacteria respond to a nanopillared surface. A variety of such surfaces have been studied over the past decade and have been shown to inhibit colonization. There are important implications to the development of infection-resisting surfaces, while avoiding antimicrobial use, for biomedical applications. Previous reports have largely attributed the inhibitory effect of nanopillars to their presumed ability to kill bacteria by mechanical piercing. This manuscript raises the question of whether that mechanism is valid, and, in particular, it uses electron microscopy - notably tomography - to make morphological studies of the nanopillar-bacteria interaction.

The microscopy, particularly the TEM and FIB tomography, provide new and insightful images that inform the question of how nanopillars interact with gram-positive and gram-negative bacteria. These results are impressive. To my knowledge, comparable image data have not been previously published.

Having said that, I find the manuscript in its present form to be cumbersome and difficult to understand. It would be a more impactful manuscript if it were substantially shortened - a lot of the experimental data presented are peripheral and can be omitted - and remained focused on its principal question(s).

Some specific comments are:

1. The title would be better if it highlighted what new knowledge has been developed rather than highlighting the techniques used.
2. Calling the surfaces "nanopillar arrays" suggests an order that simply is not present. The authors could probably omit the word array with no loss of meaning.
3. The manuscript would be substantially strengthened if there was a more clear articulation in the Introduction of the specific questions to be addressed. Microscopy to assess penetration is somewhat obvious from the focus on this topic in the preceding paragraph ("To elucidate the impact of TiO₂ nanowire arrays on bacterial envelope integrity and cell ultrastructure, electron

tomography techniques were utilised to reconstruct detailed 3D visualisations of bacteria adhered to nanowire arrays.") By extension, one can surmise why a live/dead assay would be relevant. However, the need for a proteomic analysis isn't obvious unless the reader understands that the nanopillars don't kill most of the bacteria, and this isn't revealed until much later. Notably, the 3rd paragraph of the Discussion starts reviewing the literature on oxidative stress and argues that this has been seen on other nanopillared surfaces. Doesn't this provoke a hypothesis, which could be presented in the Introduction, about a specific proteomic response?

4. The biomimetic parallel to cicada and dragon-fly wings is overdone. Mentioning it is appropriate, but the biomimetic aspects dominate the Introduction and the beginning of the Results. As interesting as they may be, the biomimetic aspects of the subject are peripheral to the main focus of the killing mechanism. Furthermore, the fact that the synthetic surfaces are highly hydrophilic while the natural surfaces are highly hydrophobic raises some ambiguity about whether these TiO₂ surfaces are really as biomimetic as the authors would like to portray them. I recommend omitting most of Figure 1, including the table. Simply presenting a representative SEM image, with a high-magnification inset, to illustrate the nanopillar morphology would be sufficient.

5. It's not clear whether studying two different nanopillared surfaces (NW-715-45 and NW-850-5) is important. Doing so simply seems to add another independent variable that is not directly connected to the basic question of whether nanopillars kill bacteria or change their physiological response. It seems that NW-850-5 was used for the TEM and FIB tomography experiments, the results of which are one (the?) key focal point of the manuscript. The authors can very likely draw the same conclusions they make now with no loss by omitting NW-714-45 altogether.

6. The SEM images in Figure 2, while very nice, do not provide enough useful information to justify the investment of space. This section is entitled: "TiO₂ nanowires induce envelope deformation and penetration." However, the paragraph following this section heading digresses to discuss the early stages of surface colonization. A lot of text is then dedicated to the effect of culture time (0.5, 1, 3, 10 h). It's not clear why this (additional) independent variable of culture time is central to the main theme of establishing of whether nanopillars kill bacteria or change their physiological response. Why were these time points chosen? A more impactful time-resolved focus would address whether nanopillar penetration affects overall biofilm formation but that question is well beyond the scope of this manuscript and certainly won't be addressed with 10 h culture studies. The section would be less cumbersome if the authors cut right to the chase with their sentence in the second paragraph (starting "On control surfaces, bacteria..."): "Nanowire-induced envelope deformation was more prominent for *E. coli* and *K. pneumoniae*,..." Figure 2 could then simply focus on the justifying this statement.

7. The manuscript in several places indicates that envelop deformation/penetration was observed at either low or high frequency. It's not clear how a "frequency analysis" was performed. It appears that the authors are making a qualitative statement based on the results of one or a few SEM images. The absence or presence of a large fraction of cells with or without envelope deformation would seem to have a bearing on a central focus of the manuscript, namely to elucidate whether killing occurs by nanopillar penetration. I recommend developing more careful language about what constitutes a deformation and then analyze a relatively large number of bacteria to quantify the frequency of deformation occurring.

8. The text associated with the scale bars on the various images is too small.

9. In the Methods (TEM sectioning and imaging) section, it's not clear if the microtomy of the resin-embedded bacteria on the nanopillared surfaces was done with the 6-4 Ti substrate in place and thus included in the resulting thin section.

10. Presumably the TEM tomography done using the Technai 20 was based on a bright-field TEM tilt series. This point should be clarified.

11. Central to the understanding of the tomography results are the words "deformation," "penetration," and "lysis." The manuscript would be strengthened if the authors clearly define what they mean by each of these terms.
12. The manuscript needs to explain what segmentation algorithm and criteria were used to create the reconstructions in, for example, figures 4e and 4f and how these images are used to differentiate between envelope deformation and penetration.
13. The text labels on the images (e.g. figures 4e, 4f, and 5h, and others) need to use a larger font.
14. I don't find Figures 5b - 5g to be useful. These can be omitted.
15. The paragraph preceding figure 6 starts with: "To even more precisely investigate the effects..." It's not clear that FIB tomography is more precise than TEM tomography, and, while minor, this phrase is ambiguous.
16. The section resolution for the FIB tomography is stated in the Methods section as 30 nm. Following from the previous comment, this is not high precision. More importantly, there must be a lot of imaging processing/averaging associated with the FIB reconstructions to not see pixelation in, for example, figure 7e, which indicates an envelope thickness of 35 nm (i.e. $35 \text{ nm} \sim 30 \text{ nm}$). A better discussion of the post processing is needed.
17. The manuscript needs to define what is meant by "RLU."

Reviewer #4 (Remarks to the Author):

Comments for the authors

Jenkins and coworkers explore the possibility to use a bio-inspired surface to counteract bacterial biofilm formation on surgical implants thus avoiding the use of antibiotic pre-treatment of such implants to control infection.

Their main objective was to establish if the biomimetic surface is efficient in inhibiting biofilm formation and microbial growth, and also to clarify some aspects concerning the reason of microbial cell death, that were poorly elucidated in previous literature data.

The authors tried to solve this challenge by focusing on dragonfly wing-mimetic TiO₂ nanowire arrays and exploring their effects on both Gram-positive and Gram-negative bacteria. The results obtained by combining several methods (SEM; TEM; 3D, Luminescence and quantitative tandem mass proteomics) highlight that the designed mimetic nanowire arrays never result in cell lysis but cause both morphological and physiological variations on the bacterial phenotypes (envelope deformation, inhibition of cell division, inhibition of colony/biofilm formation, oxidative stress response activation) in both Gram-positive and Gram-negative bacteria. The authors conclude that oxidative perturbation is the main cause of cell impairment and death induced by the nanoprotruding biomimetic arrays.

The strength of this paper lies in the combined use of several techniques to achieve a larger viewpoint on the problem to be analyzed and in the high quality of the figures unambiguously demonstrating what happens at the microscopic level. Although there are no "weak points" on the reported results, the paper could be improved and become more suitable to the journal impact factor, if analyses of protein post-translational modifications (PTMs) are performed before final publication. This would render the paper best updated and can bring overall novelty and added value to the article.

Major points

1-PROTEIN PTMs. Since 2008, the concept of protein species has been established (Jungblut et al: The speciation of the proteome). In prokaryotes, PTMs are considered one of the possible way to regulate physiology and for bacterial rapid adaptation in response to environmental changes, stressors, host factors (Cain et al, 2014, JOP). Concerning oxidative stress in particular, redox regulation via reversible thiolation (Loi et al, 2015) and hydroxylation (Van Staalduinen and Jia 2015) has been reported.

In my opinion, it is not necessary to repeat the experiments, only to analyze with a second run the characterized proteins for searching PTMs (re-interrogation of old mass spectrometry data). At present high quality informatics tools can provide robust and definitive PTMs characterization. For a good overview concerning the methods, the recent article by Gupta et al, : "Whole proteome analysis of post-translational modifications: application of mass spectrometry for proteogenomic annotation" published in 2018 on Genome Research (17, 1362-1377) will help the authors.

2- PROTEOMIC DATA SHARING. It could be useful for the scientific community if the proteomic mass spectra will be deposited to the ProteomeXchange Consortium.

Minor point.

In the Proteomic Data Analysis paragraph please add the unit of measurement (Da) to the values of oxidation of methionine, carbamidomethylation of cysteine and the TMT mass tag.

Reviewer#1 (remarks to the author):

The manuscript by Su et al. addresses the very important problem of finding basic solutions for preventing hospital-acquired infections through novel antibacterial implant surfaces. The new nanowire materials surfaces presented here are bioinspired by cicada and dragonflies nanoarrays.

The authors show convincingly how nanowire arrays induce deformation and penetration of Gram-positive and Gram-negative envelopes but do not rupture or lyse the microbes. In addition it is shown by proteomic analysis and cell test that the titanium nanowires also have capacity to impede bacterial division and trigger oxidative stress responses. The discovery of the latter mechanism can be considered groundbreaking.

The paper is very well written and the claims are supported very well by experimental evidence and a clear discussion. The experiments are very well designed and well conducted. The TEM and SEM results are very impressive. The study delivers new insights into mechanisms previously unknown this may have potential to have major impact for different fields. Therefore, the presented results are highly useful for others in the community and the wider fields of materials science and microbiology and medicine as a whole.

I fully support publication of the manuscript in Nature Communications after some minor revisions described in the following:

Author response: Thank you for your positive comments.

Reviewer#1 (suggested revisions):

1. The authors present in Supplementary Figure 2 a GXID spectrum and state that 98% of the nanowires are rutile. Yet, the other signals in the diffraction pattern are (for example alpha titanium) are very strong. This seems like a contradiction. How can this be explained?

Author response: The calculated ratio of anatase to rutile was determined using the Spurr and Myers equation, by calculating the area underneath the characteristic anatase ($2\theta = 26.5$) and rutile ($2\theta = 27.5$) peaks. The alpha titanium peaks were not included in this analysis as they relate to titanium and not titanium dioxide. We realise that this point was not clear in our original manuscript and have now revised this figure and figure legend to clearly highlight the peaks used for analysis.

2. What is the nature of the dark bands that can be seen in the nanowires in Fig. 5a? Do they indicate composition fluctuations?

Author response: The dark bands seen within nanopillars are called bend contours; these spatial contrasts are often seen in crystalline materials that are locally deformed or bent. This leads to spatial differences in the orientation of crystals and means that an incident electron beam will diffract at a different angle, depending on its incident location along the nanopillar. We refer the reviewer to Figure 6b, in which it is clear

that nanopillar 1 is not completely straight, which could lead to bend contours under bright or dark field TEM.

3. How do the presented results compare to related previous results from Dewald et al. in *Colloids and Surfaces B: Biointerfaces* 163, 201-208 and Lüdecke et al. 145 (2016) 617-625? These use related materials and therefore these related works should at least be mentioned.

Author response: We thank the reviewer for this suggestion, and have cited these publications to highlight the use of FIB-SEM as a tool for investigating the cell-material interface. However, we believe a direct comparison with Dewald et al. (2018) and Lüdecke et al. (2016) is beyond the scope of the current manuscript, since the underlying mechanism being investigated is fundamentally different (i.e. antibiofouling surface and not physical contact killing surface).

4. Can the author elaborate a bit more for which microbes the presented approach will be effective and also what its limits are (for example microbes that will not be effected).

Author response: The manuscript has been modified to now incorporate discussion of the potential limits of the nanotextured surfaces with regard to the spectrum of their antimicrobial activity.

5. Why are pili and fimbria absent for the microbes?

Author response: As indicated in Figure 2, fimbriae were evident on the surface of bacteria exposed to control surfaces, but were not seen for bacteria on the nanopillar surfaces, implying that this was a nanotopography-mediated effect. Proteomics data for *E. coli* indicated significant upregulation of GlgS in cells incubated on nanopillars relative to those on the control surface. GlgS is a negative regulator of type I fimbriae (Rahimpour et al., 2013). We therefore hypothesise that bacterial contact with our nanopillar surfaces induced downregulation of fimbrial expression and perhaps that of other surface appendages. Further investigations into these possible nanotopography-mediated effects are currently under investigation. The manuscript has been modified to better articulate this discussion point.

Reviewer #2 (remarks to the author):

The submitted article investigates the bacterial responses to titanium nanowire arrays using a range of techniques including electron tomography, biochemical assays and quantitative proteomic analysis. The authors determined the morphological and physiological responses of bacteria to dragonfly mimetic nanowire arrays, constructed 3D visualisations of bacteria adhered to nanowire arrays, determined changes in protein expression and quantified the growth and viability of *S. aureus*, *Staphylococcus epidermidis*, *E. coli* and *Klebsiella pneumoniae* populations.

The major claims of the report are as follows;

1) The nanowire arrays induce deformation and penetration of Gram-positive and Gram-negative envelopes, but do not rupture or lyse bacteria. This is a really interesting point and it is the first time that this has been demonstrated clearly. Many authors make suggestions that biomimetic or nanopillars surfaces are causing cell rupture, but do not prove it. This paper clearly does, and this is novel and will be of great interest to others in this research area and the wider research community.

2) Nanowires also have capacity to impede bacterial division and trigger oxidative stress responses. This was clearly proven. Why was this assay selected rather than others? It needs defending why only ROS was evaluated.

However, although this is an excellent manuscript, that is novel, truly multidisciplinary, well written and well thought out, there are a few points to be addressed.

The proteomics data is important and the results from this part of the work are missed in the abstract, and throughout much of the paper. These results need weaving throughout the paper, especially in terms in what they contribute to the understanding of the bacterial processes. Otherwise this looks like a separate piece of work.

The results begin with *S. epidermidis* included, then they disappear, only to be picked up later on in the manuscript. Throughout the paper, either refer to all the bacterial results or remove the bacteria that there is only partial data for.

With regards the colony forming units (CFU), it would be advantageous to know what area coverage this relates too.

The statistical analysis is included and satisfactory.

As such the manuscript should be highly considered for publication following minor corrections

Author response: Thank you for your support for this work.

Reviewer #2 (suggested revisions):

1. Nanowires also have capacity to impede bacterial division and trigger oxidative stress responses. This was clearly proven. Why was this assay selected rather than others? It needs defending why only ROS was evaluated.

Author response: The decision to investigate ROS levels was informed by the proteomics analysis, which indicated an increased abundance in *S. aureus* or *E. coli* proteins involved in protection from ROS or repair of oxidative damage. The manuscript has been modified to explain this selection more clearly. Having provided preliminary validation of the proteomics data, it is hoped that a more expansive study of potential oxidative stress responses induced by nanopillar arrays can form the basis of future studies.

2. The proteomics data is important and the results from this part of the work are missed in the abstract, and throughout much of the paper. These results need weaving throughout the paper, especially in terms in what they contribute to the understanding of the bacterial processes. Otherwise this looks like a separate piece of work.

Author response: The manuscript has been modified to include the major findings from proteomics data within the abstract. We have also amended the manuscript to highlight the importance of molecular studies and how these analyses progress our fundamental understanding of bacterial responses to nanopillar arrays.

3. The results begin with *S. epidermidis* included, then they disappear, only to be picked up later on in the manuscript. Throughout the paper, either refer to all the bacterial results or remove the bacteria that there is only partial data for.

Author response: As there is only partial data available for *S. epidermidis*, all data pertaining to *S. epidermidis* have been removed from the manuscript.

4. With regards the colony forming units (CFU), it would be advantageous to know what area coverage this relates too.

Author response: Figure 8 has been amended to include the area coverage to which the CFU data correspond.

Reviewer #3 (Remarks to the Author):

This manuscript addresses the important topic of how bacteria respond to a nanopillared surface. A variety of such surfaces have been studied over the past decade and have been shown to inhibit colonization. There are important implications to the development of infection-resisting surfaces, while avoiding antimicrobial use, for biomedical applications. Previous reports have largely attributed the inhibitory effect of nanopillars to their presumed ability to kill bacteria by mechanical piercing. This manuscript raises the question of whether that mechanism is valid, and, in particular, it uses electron microscopy - notably tomography - to make morphological studies of the nanopillar-bacteria interaction.

The microscopy, particularly the TEM and FIB tomography, provide new and insightful images that inform the question of how nanopillars interact with gram-positive and gram-negative bacteria. These results are impressive. To my knowledge, comparable image data have not been previously published.

Having said that, I find the manuscript in its present form to be cumbersome and difficult to understand. It would be a more impactful manuscript if it were substantially shortened - a lot of the experimental data presented are peripheral and can be omitted - and remained focused on its principal question(s).

Author response: We thank the reviewer for highlighting the importance of this manuscript.

Reviewer #3 (suggested revisions):

1. The title would be better if it highlighted what new knowledge has been developed rather than highlighting the techniques used.

Author response: The title has been amended to the following: “Nanopillar surface mediates antibacterial effects via cell impedance, penetration and induction of oxidative stress”.

2. Calling the surfaces "nanopillar arrays" suggests an order that simply is not present. The authors could probably omit the word array with no loss of meaning.

Author response: We have removed the term ‘array’ in reference to the nanopillared surfaces.

3. The manuscript would be substantially strengthened if there was a clearer articulation in the Introduction of the specific questions to be addressed. Microscopy to assess penetration is somewhat obvious from the focus on this topic in the preceding paragraph ("To elucidate the impact of TiO₂ nanowire arrays on bacterial envelope integrity and cell ultrastructure, electron tomography techniques were utilised to reconstruct detailed 3D visualisations of bacteria adhered to nanowire arrays.") By extension, one can surmise why a live/dead assay would be relevant. However, the need for a proteomic analysis isn't obvious unless the reader understands that the nanopillars don't kill most of the bacteria, and this isn't revealed until much later. Notably, the 3rd paragraph of the Discussion starts reviewing the literature on oxidative stress and argues that this has been seen on other nanopillared surfaces. Doesn't this provoke a hypothesis, which could be presented in the introduction, about a specific proteomic response?

Author response: We have amended the manuscript to more clearly articulate the central questions being addressed in this study, and the rationale for considering bacterial stress responses by exploiting quantitative proteomic analysis.

4. The biomimetic parallel to cicada and dragon-fly wings is overdone. Mentioning it is appropriate, but the biomimetic aspects dominate the Introduction and the beginning of the Results. As interesting as they may be, the biomimetic aspects of the subject are peripheral to the main focus of the killing mechanism. Furthermore, the fact that the synthetic surfaces are highly hydrophilic while the natural surfaces are highly hydrophobic raises some ambiguity about whether these TiO₂ surfaces are really as biomimetic as the authors would like to portray them. I recommend omitting most of Figure 1, including the table. Simply presenting a representative SEM image, with a high-magnification inset, to illustrate the nanopillar morphology would be sufficient.

Author response: The introduction has been revised to focus on the proposed killing mechanism of nanostructured materials. Figure 1 has been amended to include representative SEM images of nanopillar surface NW-815-5 only.

5. It's not clear whether studying two different nanopillared surfaces (NW-715-45 and NW-850-5) is important. Doing so simply seems to add another independent variable that is not directly connected to the basic question of whether nanopillars kill bacteria or change their physiological response. It seems that NW-850-5 was used for the TEM and FIB tomography experiments, the results of which are one (the?) key focal point of the manuscript. The authors can very likely draw the same conclusions they make now with no loss by omitting NW-714-45 altogether.

Author response: NW-715-45 has been removed from the manuscript.

6. The SEM images in Figure 2, while very nice, do not provide enough useful information to justify the investment of space. This section is entitled: "TiO₂ nanowires induce envelope deformation and penetration." However, the paragraph following this section heading digresses to discuss the early stages of surface colonization. A lot of text is then dedicated to the effect of culture time (0.5, 1, 3, 10 h). It's not clear why this (additional) independent variable of culture time is central to the main theme of establishing of whether nanopillars kill bacteria or change their physiological response. Why were these time points chosen? A more impactful time-resolved focus would address whether nanopillar penetration affects overall biofilm formation but that question is well beyond the scope of this manuscript and certainly won't be addressed with 10 h culture studies. The section would be less cumbersome if the authors cut right to the chase with their sentence in the second paragraph (starting "On control surfaces, bacteria..."): "Nanowire-induced envelope deformation was more prominent for *E. coli* and *K. pneumoniae*,..." Figure 2 could then simply focus on the justifying this statement.

Author response: As highlighted on page 5 (lines 25-28), previous studies had shown significant changes in bacterial morphology within 3 hours of contact with nanopillar surfaces. The incubation times used in this study were therefore selected to enable direct comparison with this prior work. We have revised our manuscript to provide greater clarity on this time point selection. SEM images for 10 h have been moved to Supplementary Figure 5. Figure 2 now focuses on justifying envelope deformation in *S. aureus*, *E. coli* and *K. pneumoniae* observed at 3 h.

7. The manuscript in several places indicates that envelope deformation/penetration was observed at either low or high frequency. It's not clear how a "frequency analysis" was performed. It appears that the authors are making a qualitative statement based on the results of one or a few SEM images. The absence or presence of a large fraction of cells with or without envelope deformation would seem to have a bearing on a central focus of the manuscript, namely to elucidate whether killing occurs by nanopillar penetration. I recommend developing more careful language about what constitutes a deformation and then analyze a relatively large number of bacteria to quantify the frequency of deformation occurring.

Author response: We have modified the language to indicate the frequency of nanopillar induced envelope deformation and penetration and amended the methodology to explain how these were determined.

8. The text associated with the scale bars on the various images is too small.

Author response: All figure scale bars have been enlarged.

9. In the Methods (TEM sectioning and imaging) section, it's not clear if the microtomy of the resin-embedded bacteria on the nanopillared surfaces was done with the 6-4 Ti substrate in place and thus included in the resulting thin section.

Author response: We have revised this methods section to more clearly state that microtomy of the resin-embedded bacteria on the nanopillared surfaces was performed while bacteria were in place on the Ti-64 substrates.

10. Presumably the TEM tomography done using the Technai 20 was based on a bright-field TEM tilt series. This point should be clarified.

Author response: We have amended the methodology section to more clearly indicate that TEM tomography was based on a tilt series performed in bright-field mode.

11. Central to the understanding of the tomography results are the words "deformation," "penetration," and "lysis." The manuscript would be strengthened if the authors clearly define what they mean by each of these terms.

Author response: Bacterial envelope deformation, penetration and cell lysis have now been more clearly defined.

12. The manuscript needs to explain what segmentation algorithm and criteria were used to create the reconstructions in, for example, figures 4e and 4f and how these images are used to differentiate between envelope deformation and penetration.

Author response: Further information regarding image segmentation and reconstruction has been included in the methods section. More information is also provided regarding how envelope deformation and penetration are distinguished, based on the definitions provided in this manuscript.

13. The text labels on the images (e.g. figures 4e, 4f, and 5h, and others) need to use a larger font.

Author response: All text labels have been enlarged.

14. I don't find Figures 5b - 5g to be useful. These can be omitted.

Author response: Figures 5b – 5g have been removed.

15. The paragraph preceding figure 6 starts with: "To even more precisely investigate the effects..." It's not clear that FIB tomography is more precise than TEM tomography, and, while minor, this phrase is ambiguous.

Author response: This section has been revised to the following: "To investigate the effects of nanopillars on bacterial envelope morphology further...".

16. The section resolution for the FIB tomography is stated in the Methods section as 30 nm. Following from the previous comment, this is not high precision. More importantly, there must be a lot of imaging processing/averaging associated with the FIB reconstructions to not see pixelation in, for example, figure 7e, which indicates an envelope thickness of 35 nm (i.e. 35 nm ~ 30 nm). A better discussion of the post processing is needed.

Author response: Although the section thickness was 30 nm (i.e. the distance between consecutive ion beam sections), the resolution of SEM images used for 3D reconstructions was approximately 1-2 nm. We have amended the manuscript to more clearly indicate the post processing steps taken to reconstruct the models from consecutive SEM images.

17. The manuscript needs to define what is meant by "RLU."

Author response: The manuscript has been revised to define relative luminescence units (RLU).

Reviewer #4 (Remarks to the Author):

Jenkins and coworkers explore the possibility to use a bio-inspired surface to counteract bacterial biofilm formation on surgical implants thus avoiding the use of antibiotic pre-treatment of such implants to control infection.

Their main objective was to establish if the biomimetic surface is efficient in inhibiting biofilm formation and microbial growth, and also to clarify some aspects concerning the reason of microbial cell death, that were poorly elucidated in previous literature data.

The authors tried to solve this challenge by focusing on dragonfly wing-mimetic TiO₂ nanowire arrays and exploring their effects on both Gram-positive and Gram-negative bacteria. The results obtained by combining several methods (SEM; TEM; 3D, Luminescence and quantitative tandem mass proteomics) highlight that the designed mimetic nanowire arrays never result in cell lysis but cause both morphological and physiological variations on the bacterial phenotypes (envelope deformation, inhibition of cell division, inhibition of colony/biofilm formation, oxidative stress response activation) in both Gram-positive and

Gram-negative bacteria. The authors conclude that oxidative perturbation is the main cause of cell impairment and death induced by the nanoprotruding biomimetic arrays.

The strength of this paper lies in the combined use of several techniques to achieve a larger viewpoint on the problem to be analyzed and in the high quality of the figures unambiguously demonstrating what happens at the microscopic level. Although there are no “weak points” on the reported results, the paper could be improved and become more suitable to the journal impact factor, if analyses of protein post-translational modifications (PTMs) are performed before final publication. This would render the paper best updated and can bring overall novelty and added value to the article.

Author response: We thank the reviewer for highlighting the strength of this manuscript.

Reviewer #4 (suggested revisions):

1. PROTEIN PTMs. Since 2008, the concept of protein species has been established (Jungblut et al: The speciation of the proteome). In prokaryotes, PTMs are considered one of the possible way to regulate physiology and for bacterial rapid adaptation in response to environmental changes, stressors, host factors (Cain et al, 2014, JOP). Concerning oxidative stress in particular, redox regulation via reversible thiolation (Loi et al, 2015) and hydroxylation (Van Staalduinen and Jia 2015) has been reported. In my opinion, it is not necessary to repeat the experiments, only to analyze with a second run the characterized proteins for searching PTMs (re-interrogation of old mass spectrometry data). At present high quality informatics tools can provide robust and definitive PTMs characterization. For a good overview concerning the methods, the recent article by Gupta et al: “Whole proteome analysis of post-translational modifications: application of mass spectrometry for proteogenomic annotation” published in 2018 on Genome Research (17, 1362-1377) will help the authors.

Author response: We thank the reviewer for this valuable suggestion and agree that it could be insightful to explore the PTMs of the characterised proteins. However, we believe this to be beyond the scope of the current manuscript.

2. PROTEOMIC DATA SHARING. It could be useful for the scientific community if the proteomic mass spectra will be deposited to the ProteomeXchange Consortium.

Author response: All proteomic data has been deposited to PRIDE via the ProteomeXchange submission tool.

3. In the Proteomic Data Analysis paragraph please add the unit of measurement (Da) to the values of oxidation of methionine, carbamidomethylation of cysteine and the TMT mass tag.

Author response: Units (Da) have been added to the values of oxidation of methionine and carbamidomethylation of cysteine and the TMT mass tag.

REVIEWERS' COMMENTS:

Reviewer #1 (Remarks to the Author):

All queries have been answered to my fullest satisfaction. I recommend publication in this current revised form with high priority. Congratulations to the authors on an excellent manuscript.

Reviewer#1 (remarks to the author):

All queries have been answered to my fullest satisfaction. I recommend publication in this current revised form with high priority. Congratulations to the authors on an excellent manuscript.

Author response: Thank you for your positive comments.